# Solid solution for catalytic ammonia synthesis from nitrogen and hydrogen gases at 50 °C

Masashi Hattori [1], Shinya Iijima[1], Takuya Nakao[2], Hideo Hosono [2✉] & Michikazu Hara [1✉]

The lack of efficient catalysts for ammonia synthesis from $N_2$ and $H_2$ gases at the lower temperature of ca. 50 °C has been a problem not only for the Haber–Bosch process, but also for ammonia production toward zero $CO_2$ emissions. Here, we report a new approach for low temperature ammonia synthesis that uses a stable electron-donating heterogeneous catalyst, cubic CaFH, a solid solution of $CaF_2$ and $CaH_2$ formed at low temperatures. The catalyst produced ammonia from $N_2$ and $H_2$ gases at 50 °C with an extremely small activation energy of 20 kJ mol$^{-1}$, which is less than half that for conventional catalysts reported. The catalytic performance can be attributed to the weak ionic bonds between $Ca^{2+}$ and $H^-$ ions in the solid solution and the facile release of hydrogen atoms from $H^-$ sites.

[1] Laboratory for Materials and Structures, Tokyo Institute of Technology, 4259 Nagatsuta, Midori-ku, Yokohama 226–8503, Japan. [2] Materials Research Center for Element Strategy, Tokyo Institute of Technology, 4259 Nagatsuta, Midori-ku, Yokohama, Kanagawa 226–8503, Japan. ✉email: hosono@msl.titech.ac.jp; mhara@msl.titech.ac.jp

The Haber–Bosch process currently enables the provision of food for over 70% of the world's population, consuming 2% of global energy and generating 3% of global $CO_2$ emissions[1,2]. These values would steeply increase by a rapid increase of the human population. Highly efficient conversion of $N_2$ and $H_2$ to ammonia with low-energy consumption has remained a challenge since the creation of the Haber–Bosch process, where sustainable ammonia production using natural energy is the ultimate goal. Ammonia is equilibrated in $N_2$ and $H_2$. Figure 1a demonstrates the correlation of the theoretical ammonia yield with reaction pressure and temperature. As the reaction temperature increases, ammonia decomposition as an endothermic reaction exceeds ammonia formation, an exothermic reaction, which decreases the ammonia yield. For this reason, the reaction system must therefore be further pressurized with an increase in the reaction temperature to obtain the same ammonia yield; high reaction temperature causes high pressurization, which requires large energy consumption for both heating and pressurization. The iron-based catalysts used in the present Haber–Bosch process are effective for ammonia synthesis above 350 °C, so that the maximum ammonia yield is at most 30−40%, despite excess pressurization (>10–20 MPa) accompanied by large energy consumption. This is a serious drawback in sustainable ammonia production without the use of fossil fuels. While wind power generation has been reported to be compatible with the Haber–Bosch process, the process itself consumes 40–50% of the electric power generated by wind turbine, and thereby electric power available for $H_2$ production is considerably limited[3]. Figure 1a also indicates that the ammonia yield exceeds 98% at ca. 50 °C regardless of pressures, and there is no significant difference in ammonia yield among pressures below this temperature. Thus, a lower temperature is favorable for ammonia production with respect to yield and energy consumption, and more efficient ammonia production is required to overcome the kinetic barrier at lower temperature to achieve the equilibrium. However, conventional catalysts equally lose the catalytic activity for ammonia formation from $N_2$ and $H_2$ at 100–200 °C, even if they exhibit high catalytic performance at high temperatures, as shown in Fig. 1a. Lowering the temperature for a loss of activity below 50 °C would largely enhance the catalytic activity for ammonia synthesis at low-temperature range below 300 °C. While there has been significant progress in homogeneous catalytic systems to synthesize ammonia from $N_2$ and $H^+$ activated by specific and nonreusable reagents below room temperature[4,5], guiding principles to lower the temperature for a loss of activity on ammonia synthesis from $N_2$ and $H_2$ have yet to be clarified.

Thus, the lack of catalysts that are workable at lower temperatures has remained a problem for the Haber–Bosch process for over a century, and has also prevented sustainable ammonia production toward zero $CO_2$ emissions. We have begun to re-examine the low-temperature kinetics of ammonia synthesis catalysts to find a route for low-temperature ammonia production. Figure 1b shows Arrhenius plots for a commercial Fe catalyst[6] with ammonia formation rates that were both measured ($r^M NH_3$) and estimated from the Arrhenius equation ($r^E NH_3$). The reaction rate follows the Arrhenius equation as long as the reaction mechanism is unchanged in the temperature range; therefore, the reaction rate at a specific temperature was estimated using the Arrhenius equation. The difference between $r^M NH_3$ and $r^E NH_3$ increases with a decrease in the temperature below 300 °C. The Arrhenius equation predicted a sufficient amount of ammonia to form at 100 °C. However, no ammonia formation was detected below 150 °C, which brought the natural logarithm of the rate close to −∞. Even if the catalyst amount and space velocity were increased significantly, ammonia formation was not observed below 150 °C. Taking into account the detection sensitivity of the ammonia analysis methods, the rate of ammonia formation was expected to be less than nano mol $h^{-1} g^{-1}$. This means that the catalyst cannot act for ammonia synthesis at all below the temperature. The same phenomenon was confirmed in several representative catalytic systems for the synthesis of ammonia from $N_2$ and $H_2$ (Supplementary Table 1). This cannot be simply attributed to deactivation by ammonia adsorbed on the catalyst, because ammonia that adsorbs on transition metals (TMs) such as Fe and Ru will desorb below room temperature[7]. Ammonia formation from $N_2$ and $H_2$ over catalysts proceeds through the dissociative adsorption of $N_2$ ($N_2 \rightarrow 2 N$), followed by the hydrogenation of nitrogen adatoms ($N \rightarrow NH_3$). The former step has a higher energy barrier than the latter that proceeds on TM surfaces at room temperature to 100 °C[8]. It is well-known that the cleavage of $N_2$ molecules with ammonia synthesis catalysts is largely enhanced by electron donation from electron-donating materials into the $\pi^*$ orbitals of $N \equiv N$ via the d-orbitals of TMs[9–11], and the electron-donating capability of the TM itself is almost independent of the temperature below 200 °C[12,13] (see Supplementary Discussion). A decrease in the electron-donating capability has been purported as one possible explanation for the lack of ammonia synthesis by catalysis at low temperatures.

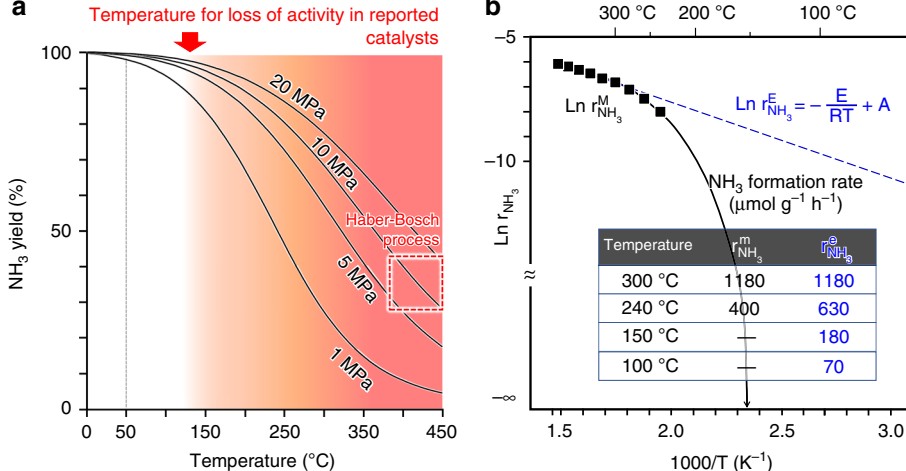

**Fig. 1 Ammonia synthesis from $N_2$ and $H_2$. a** Correlation of ammonia yield with temperature and pressure. **b** Arrhenius plots for ammonia synthesis over a commercial Fe catalyst.

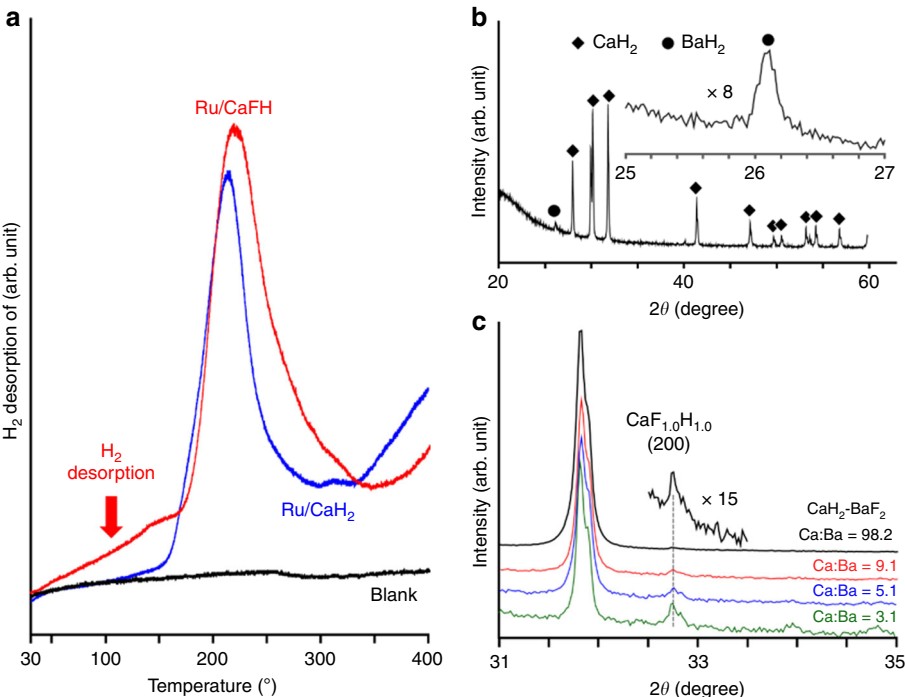

**Fig. 2 H$_2$-TPD and XRD for prepared materials. a** H$_2$-TPD profiles for Ru/CaH$_2$ and Ru/CaFH after ammonia synthesis reaction at 340 °C, followed by cooling down below 20 °C. TPD measurements were performed under Ar flow (1 °C min$^{-1}$). **b** XRD pattern for CaH$_2$–BaF$_2$ mixture (Ca:Ba =98:2) after heating at 340 °C for 10 h in H$_2$. **c** Narrow-range XRD patterns for CaH$_2$–BaF$_2$ mixtures (Ca:Ba = 98:2, 9:1, 5:1, and 3:1) after heating at 340 °C for 10 h in H$_2$.

Therefore, stable materials that exhibit high electron-donating capability at low temperatures may lead to the realization of low-temperature ammonia synthesis.

Here we present a new approach for low-temperature ammonia synthesis that uses a stable electron-donating heterogeneous catalyst, Ru nanoparticle-deposited cubic CaFH solid solution. The catalyst produces ammonia from N$_2$ and H$_2$ gases at 50 °C with an extremely small activation energy of 20 kJ mol$^{-1}$, which is less than half that for conventional catalysts reported. The catalytic performance can be attributed to the weak ionic bonds between Ca$^{2+}$ and H$^-$ ions in the solid solution and the facile release of hydrogen atoms from H$^-$ sites.

## Results

**CaFH solid solution as a strong electron-donating material.** As a first step to verify the working hypothesis and to prepare such electron-donating materials, we have focused on calcium hydride (CaH$_2$), a familiar dehydrating agent, because of its simplicity. TM nanoparticles deposited on CaH$_2$ abstract H atoms from the near-surface CaH$_2$ due to substantial interaction between the TM and H$^-$, and the H atoms move on to the metal nanoparticles and desorb as H$_2$ molecules, leaving electrons in the H$^-$ vacancy of CaH$_2$ (CaH$_2 \rightarrow$ Ca$^{2+}$H$^-_{(2-x)}$e$^-_x$ + $x$H)[14]. The resulting Ca$^{2+}$H$^-_{(2-x)}$e$^-_x$ behaves as a stable surface electride with a small work function ($\Phi$ = 2.7 eV) comparable with that of metallic Li[15]. The strong electron donation from Ca$^{2+}$H$^-_{(2-x)}$e$^-_x$ to the TM nanoparticles enhances the cleavage of N$_2$ molecules, which leads to high catalytic performance for ammonia synthesis[14]. Supplementary Table 1 shows that the difference between the r$^M$NH$_3$ and r$^E$NH$_3$ rates for the formation of ammonia over Ru-deposited CaH$_2$ (Ru/CaH$_2$) was not negligible at ≤ 200 °C, as with other catalysts; however, the catalyst was not at all active for ammonia formation at 150 °C. The temperature that eliminates the activity of Ru/CaH$_2$ is therefore between 150 and 200 °C, which is similar to that for conventional

catalysts (Supplementary Table 1). A H$_2$-temperature-programmed desorption (TPD) profile for Ru/CaH$_2$ (Fig. 2a) revealed that the H$_2$ desorption-onset temperature was almost identical to the temperature where the catalytic activity of Ru/CaH$_2$ is lost. Formation of a strong electron-donating material is a determinant for ammonia formation over Ru/CaH$_2$ at low temperatures. Therefore, lowering of the onset temperature for H$_2$ desorption would lead to a catalytic system for ammonia synthesis at lower temperatures.

Here, we have adopted a new strategy based on classical theory to lower the electride formation temperature: the introduction of F$^-$ anions into CaH$_2$. F$^-$ is an extremely hard base in hard and soft acids and bases (HSAB), and the Ca–F ionic bond (529 kJ mol$^{-1}$) is harder than the Ca–H bond (224 kJ mol$^{-1}$)[16]. Replacing a part of H$^-$ in CaH$_2$ with F$^-$ would weaken ionic bonds between Ca$^{2+}$ and H$^-$, thereby lowering the temperature for the release of H atoms from the material. This replacement would increase the energy of electrons trapped at H$^-$ vacancies due to electron repulsion between the electron and F$^-$, which would cause a reduction in the work function of the surface region and in turn enhance the electron-donating power. In this study, F$^-$ was introduced into CaH$_2$ by heating a simple mixture of CaH$_2$ and BaF$_2$ powders in a flow of H$_2$ at 340 °C. A wide-range X-ray diffraction (XRD) pattern (Fig. 2b) of the resultant sample (Ca/Ba atomic ratio = 98:2) indicated that the sample is mainly composed of CaH$_2$. Diffraction peaks of BaF$_2$ or CaF$_2$ were not observed in the XRD pattern, whereas the diffraction peaks due to BaH$_2$ were apparent (Supplementary Fig. 1). Consequently, CaF$_2$ is not formed in the heated mixture, despite the complete replacement of F$^-$ in BaF$_2$ with H$^-$ derived from CaH$_2$. CaH$_2$–BaF$_2$ mixtures in various CaH$_2$/BaF$_2$ ratios were heated in H$_2$ to identify the material formed in the heated CaH$_2$–BaF$_2$ mixture. Figure 2c shows narrow-range XRD patterns ($2\theta$ = 31–35°) of heated CaH$_2$–BaF$_2$ mixtures (Ca/Ba atomic ratios of 98:2, 9:1, 5:1, and 3:1), where an asymmetrical diffraction assignable to (200) of the cubic CaF$_{1.0}$H$_{1.0}$ solid solution appears

at $2\theta = 32.7°$[17]. It is well-known that orthorhombic $CaH_2$ is transformed into a cubic structure in the formation of cubic CaFH solid solution[17]. The diffraction peak intensity increased, but was not shifted with an increase in the Ba content. In order to clarify the characteristics of $CaF_{1.0}H_{1.0}$ solid solution formed on $CaH_2$–$BaF_2$ mixtures, $CaF_xH_{2-x}$ solid solution (denoted as $CaF_xH_{2-x}$–$CaF_2$) was prepared by heating mixtures of orthorhombic $CaH_2$ and cubic $CaF_2$ at 550 °C, a conventional method[17]. XRD patterns of $CaF_xH_{2-x}$–$CaF_2$ ($1.0 \leq x \leq 1.6$) (Supplementary Fig. 2) elucidated that the (200) diffraction for the cubic $CaF_xH_{2-x}$–$CaF_2$ is sensitive to the $F^-$ concentration and shifts from $2\theta = 32.7°$ to lower angles with decreasing $F^-$ concentration[17]. These results suggest that heating a $CaH_2$–$BaF_2$ mixture at 340 °C forms the most stable $CaF_{1.0}H_{1.0}$ on $CaH_2$. In X-ray photoelectron spectroscopy (XPS) measurements for heated $CaH_2$–$BaF_2$ mixtures (Ca/Ba atomic ratio of 98:2), $CaF_xH_{2-x}$–$CaF_2$ ($x = 1$) and $CaH_2$ (Supplementary Fig. 3), the Ca 2p peaks for both heated $CaH_2$–$BaF_2$ mixtures and $CaF_xH_{2-x}$–$CaF_2$ ($x = 1$) appeared at 346.5 eV, which is lower than that for $CaH_2$ (347.3 eV) and supports the formation of the $CaF_{1.0}H_{1.0}$ solid solution on the surface of a small amount of $BaF_2$-added $CaH_2$. Next, Ru nanoparticles (12 wt%) were deposited on $CaF_{1.0}H_{1.0}$ phase obtained by heating a $CaH_2$–$BaF_2$ (Ca:Ba=98:2) mixture at 340 °C (denoted as Ru/CaFH), and Ru/CaFH after ammonia synthesis reaction over 30 h at 340 °C was examined by $H_2$-TPD (Fig. 2a). The starting temperature for $H_2$ desorption was lowered to the range of room temperature to 50 °C, compared with that for Ru/$CaH_2$. Supplementary Fig. 4 is the $H_2$-TPD profile of the Ru-deposited $CaF_xH_{2-x}$–$CaF_2$ (denoted as Ru/$CaF_xH_{2-x}$–$CaF_2$ ($x = 1$)), which also indicates that $H_2$ begins to desorb from the material at ca. 50 °C. This implies that the formation of the CaFH solid solution, i.e., the formation of $Ca^{2+}-F^-$ ionic bonds, clearly weakens the $Ca^{2+}-H^-$ bond and lowers the temperature for the hydrogen release reaction. A notable feature in the $H_2$-TPD profile for Ru/CaFH is that it overlaps the $H_2$-TPD profiles of Ru/$CaH_2$ (Fig. 2a) and Ru/$CaF_xH_{2-x}$–$CaF_2$ ($x = 1$) (Supplementary Fig. 4). As a result, the CaFH solid solution may coexist with $CaH_2$ on the surface of Ru/CaFH.

To evaluate the electron-donating capability of CaFH, the work function of CaFH with $H^-$ vacancies was estimated by density-functional theory (DFT) computations ("Methods", Supplementary Fig. 5). The work function on the most stable surface (111) of CaFH calculated by DFT was 2.2 eV, which is smaller than that (2.7 eV) for $CaH_2$ with $H^-$ defects because electron repulsion between electrons and $F^-$ increases the energy of electrons trapped at $H^-$ vacancies. These results indicate that the abstraction of H atoms from CaFH forms a strong electron-donating material, which has a work function comparable to that of metallic potassium ($\Phi = 2.3$ eV).

Morphological information for Ru/CaFH is summarized in Supplementary Fig. 6. Ru/CaFH with a surface area of 30 $m^2\,g^{-1}$ consists of irregular-shaped particles of 0.5–3 μm in diameter. The particle size of Ru deposited on CaFH was estimated to be 3–4 nm. XPS measurements for F 1 s and Ca 2p revealed that the surface atomic ratio of F to Ca (F/Ca) in Ru/CaFH was 0.08, which is much smaller than that expected from $CaF_{1.0}H_{1.0}$, and also supports the coexistence of CaFH and $CaH_2$.

**Catalytic performance for ammonia synthesis at low temperatures**. The catalytic performance for ammonia formation from $N_2$ and $H_2$ for tested catalysts is summarized in Fig. 3. Figure 3 also contains the results for $CaH_2$ (Ru/$CaH_2$), a $BaH_2$–BaO mixture (Ru/$BaH_2$–BaO)[18], and Cs-doped MgO (Cs–Ru/MgO)[6,10] loaded with Ru nanoparticles, and a commercial Fe catalyst[6] for comparison. Cs–Ru/MgO has a higher

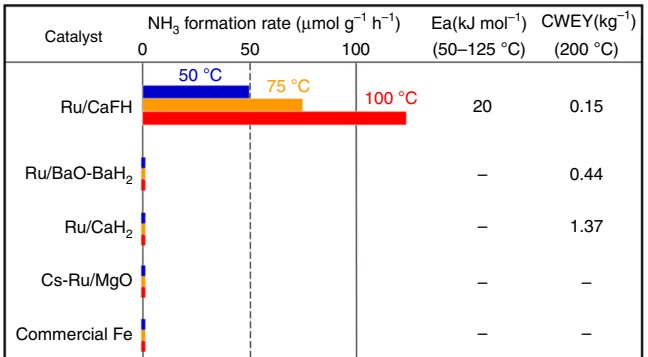

**Fig. 3 Catalytic performance.** The amounts of loaded Ru on Ru/CaFH, Ru/BaO-BaH$_2$, Ru/CaH$_2$, and Cs-Ru/MgO were 12, 10, 10, and 10 wt%, respectively. The catalytic activities of these catalysts at ca. 300 °C reached the maxima with each Ru loading. The rates of ammonia formation for Ru/CaFH at 50, 75, 100, and 125 °C were 50, 75, 120, and 190 μmol g$^{-1}$ h$^{-1}$, respectively. CWEYs were estimated from the rates of ammonia formation at each reaction temperature. It was confirmed that the rates of ammonia formation (μmol h$^{-1}$) for Ru/CaFH, Ru/BaO-BaH$_2$, and Ru/CaH$_2$ increased in direct ratio to catalyst weight (0.05–5.00 g).

catalytic activity for ammonia synthesis than the commercial Fe catalyst above 300–400 °C. Ru/$BaH_2$–BaO acts as a highly active ammonia synthesis catalyst[18] that is comparable with Ru nanoparticles immobilized on $Ca(NH_2)_2$ containing $Ba^{2+}$ (Ru/Ba–Ca $(NH_2)_2$), which exhibits the highest catalytic performance for ammonia synthesis among the reported catalysts over wide temperature (200–450 °C) and pressure ranges (0.1–0.9 MPa)[6]. Supplementary Table 2 shows the physicochemical information (surface area, porosity, and Ru particle size) and the rates of ammonia formation for Ru/CaFH, Ru/$CaH_2$, Ru/Ba–Ca$(NH_2)_2$, Ru/BaO–$BaH_2$, Ru/C12A7[19], Cs–Ru/MgO, and a commercial Fe catalyst as benchmark catalysts at 100–340 °C. It was confirmed that the catalytic activities of the commercial Fe catalyst and Cs–Ru/MgO benchmark catalysts were comparable with those reported by other groups[19–21]. These conventional catalysts did not exhibit activity for ammonia synthesis below 100–200 °C, whereas Ru/CaFH synthesized ammonia at 50 °C. The ammonia formation over the catalyst at 50 °C was confirmed by both direct mass spectrometry and ion chromatography, and is not derived from N species formed on the catalyst during the catalyst activation at 340 °C (see Supplementary Discussion). The ammonia formation rate of Ru/CaFH increased with the temperature and was unchanged even after the rate measurement was repeated, which indicates that Ru/CaFH is a stable catalyst. There was no significant difference in XRD pattern between Ru/CaFH after reaction and $CaH_2$–$BaF_2$ mixture heated at 340 °C (Ca/Ba atomic ratio = 98:2, Fig. 2b, c). The apparent activation energy for ammonia synthesis over the catalyst in the range of 50–150 °C was estimated to be 20 kJ $mol^{-1}$, which is less than half that of reported catalysts (from 40 kJ $mol^{-1}$)[6,14,18,19,21]. Furthermore, Ru/CaFH as a stable catalyst surpasses conventional catalysts at higher temperatures. Figure 3 gives the catalyst weight required for the equilibrium yield (CWEY) of ammonia at 200 °C (see Supplementary Discussion). The rates of ammonia formation and CWEYs for all tested catalysts, including Ru/Ba–Ca$(NH_2)_2$, and the recently reported highly active catalysts at 200–350 °C are summarized in Supplementary Table 3[22–25]. Although Ru/Ba–Ca $(NH_2)_2$ and Ru/BaO–$BaH_2$ have had much smaller CWEYs among the reported highly active catalysts, the CWEY of Ru/CaFH was only half that of both catalysts at 200 °C. In the case of Ru/CaFH, the apparent activation energy ($Ea = 20$ kJ $mol^{-1}$)

and coefficient due to the collision frequency ($A = 10$) in the Arrhenius equation estimated from the rates of ammonia formation at 50, 75, 100, and 125 °C were almost the same as those ($Ea = 23$ kJ mol$^{-1}$, $A = 12$) obtained by the ammonia formation rates at 275–340 °C. In addition, there was no significant difference in Ea and A for ammonia formation over Ru/CaFH at 0.1 and 0.9 MPa. This suggests that the same active sites on Ru/CaFH form ammonia through a reaction mechanism in a wide range of reaction conditions ($\geq 50$ °C, $\geq 0.1$ MPa). It was also confirmed in ammonia synthesis (240–400 °C) over a commercial Fe catalyst used in this study that Ea and A at 0.1 MPa are identical with those at 0.9 MPa[6]. Furthermore, Ru/CaFH produced ammonia without a decrease in activity for long periods of time and at higher temperatures (200 and 340 °C) (Supplementary Figs. 7 and 8). Ammonia formation over Ru/CaFH was close to the equilibrium yield, even at ca. 300 °C, because of the high catalytic performance. As a result, ammonia synthesis over Ru/CaFH in Supplementary Fig. 8 reaches the equilibrium. Despite such equilibrium conversion (i.e., catalyst deactivation test conditions), the rate of ammonia formation over Ru/CaFH was constant for over 100 h. The amount of ammonia produced by Ru/CaFH at 340 °C exceeded the amount of the used catalyst (ca. 24 mmol) within 100 min. The XRD pattern and surface atomic ratio of F to Ca (F/Ca = 0.08) for Ru/CaFH were unchanged after reaction for 100 h, which was consistent with the lack of F species such as HF detected during the reaction. These results are clearly indicative of the stability of the Ru/CaFH catalyst.

The surface areas of the supports and ammonia formation rates (340 °C) of Ru/CaFH, Ru-deposited BaH$_2$ (Ru/BaH$_2$), and Ru/CaF$_x$H$_{2-x}$–CaF$_2$ ($x = 1$) where Ru nanoparticles are deposited on the CaF$_{1.0}$H$_{1.0}$ solid solution formed by heating mixtures of orthorhombic CaH$_2$ and cubic CaF$_2$ at 550 °C are summarized in Supplementary Table 4. BaH$_2$ and CaF$_{1.0}$H$_{1.0}$ solid solution are expected to be formed on Ru/CaFH, and either or both of them can contribute to the catalytic performance of Ru/CaFH. However, Ru/BaH$_2$ had a much smaller catalytic activity for ammonia synthesis than Ru/CaF$_x$H$_{2-x}$–CaF$_2$ ($x = 1$), which indicates that the catalysis of Ru/CaFH is derived from the CaF$_{1.0}$H$_{1.0}$ solid solution. Supplementary Table 4 also shows Ru/CaF$_x$H$_{2-x}$–CaF$_2$ ($x = 1$) to be inferior to Ru/CaFH with respect to ammonia synthesis. This can be attributed to the conventional preparation method for the CaF$_{1.0}$H$_{1.0}$ solid solution. Ru/CaF$_x$H$_{2-x}$–CaF$_2$ ($x = 1$) was prepared by the deposition of Ru nanoparticles. CaF$_x$H$_{2-x}$–CaF$_2$ ($x = 1$) synthesized by high-temperature solid-state reaction at $\geq 550$ °C for 20 h and the resultant surface area was very small (1 m$^2$ g$^{-1}$), which limits the catalytic activity of Ru/CaF$_x$H$_{2-x}$–CaF$_2$ ($x = 1$). On the other hand, in the case of CaFH prepared by the new method, CaFH solid solution was formed on the surface with a surface area of 10 m$^2$ g$^{-1}$. This is considered to be the reason for the difference in activity between Ru/CaFH and Ru/CaF$_x$H$_{2-x}$–CaF$_2$ ($x = 1$).

**Electron-donating capability and reaction mechanism of Ru/CaFH.** Ammonia synthesis from N$_2$ and D$_2$ over Ru/CaFH was examined to clarify the reaction mechanism. Ru/CaFH prepared at 340 °C in a flow of H$_2$ alone was cooled down from 340 °C to 180 °C in a flow of Ar, and then N$_2$–D$_2$ was passed into Ru/CaFH at temperature under atmospheric pressure (see "Methods"). The experimental reaction time profiles for ammonia synthesis from N$_2$ and D$_2$ over Ru/CaFH are shown in Fig. 4. Soon after an increase in the $m/z = 17$ signal (NH$_3$ and NDH as fragments of ND$_2$H and NDH$_2$), the signal of $m/z = 18$ (NDH$_2$ and ND$_2$ as a fragment of ND$_3$) increased. The $m/z = 19$ (ND$_2$H) signal was observed after ca. 3 min from the introduction of N$_2$–D$_2$. The fragment ratio of $m/z = 17$ (NH$_3$), 16 (NH$_2$), and 15 (NH) in

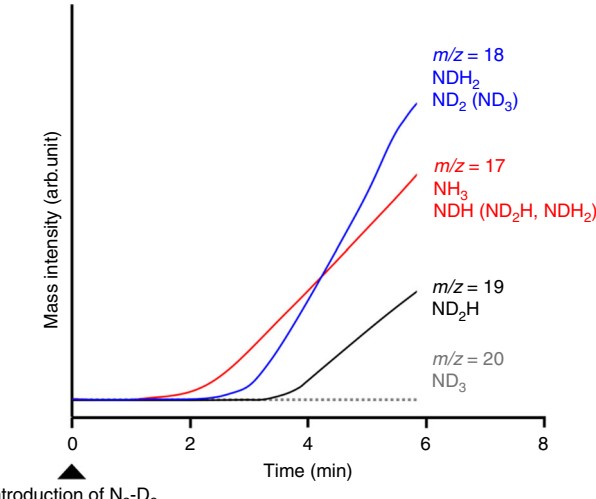

**Fig. 4 Time courses of ammonia synthesis from N$_2$ and D$_2$ over Ru/CaFH.** Reaction time profiles at the early stage of ammonia synthesis from N$_2$ and D$_2$ over Ru/CaFH at 180 °C.

pure NH$_3$ was ca. 100:80:8, so that each fragment intensity did not exceed the parent intensity. The formation of ND$_3$ ($m/z = 20$) was not observed within 10 min from the beginning of the reaction. As a result, NH$_3$ was first formed, followed by the formation of NDH$_2$ and ND$_2$H; ammonia species containing H was formed at the early stage of ammonia synthesis from N$_2$ and D$_2$ over Ru/CaFH. These results indicate that H in the CaFH bulk is used for ammonia formation at the early stage of reaction, and H can move from CaFH bulk to the Ru surface to react with N adatoms. The same phenomenon had been observed on Ru nanoparticle-deposited Ca$_2$NH (calcium nitride hydride)[14].

XPS Ru 3p$_{3/2}$ of Ru/CaFH was used to evaluate the electron-donating capability from CaFH to Ru (Supplementary Fig. 9). Ru 3p$_{3/2}$ for Ru/CaFH appeared at a slightly lower binding energy than that for metallic Ru particles deposited on SiO$_2$ (Ru/SiO$_2$), which indicates that Ru on CaFH is more negative than Ru on SiO$_2$. However, the difference was not so large, because XPS reflects only Ru atoms near the edges of Ru particles connected to CaFH from the point of view of the escape depth of photoelectrons. For this reason, we have adopted Fourier transform- infrared (FT-IR) spectroscopy measurements using N$_2$ as a probe molecule, which is a more sensitive method (FT-IR, Fig. 5). Figure 5a measured at 25 °C showed N$\equiv$N stretching ($\nu$N$_2$) bands of N$_2$ adsorbed on Ru/CaH$_2$ at 2100–2250 cm$^{-1}$, lower than that of gaseous N$_2$ (2744 cm$^{-1}$), which indicated that the electron donation from the catalyst to the antibonding $\pi$* orbitals of adsorbed N$_2$ via Ru d-orbitals (i.e., back donation) weakens the N$\equiv$N bond. The $\nu$N$_2$ bands have been reported to appear at 2100–2300 cm$^{-1}$ in highly active catalysts for ammonia synthesis[19,26]. However, these catalysts cannot function at low temperatures; such electron donation capability is insufficient to realize low-temperature ammonia synthesis. In the case of Ru/CaFH, $\nu$N$_2$ was observed in the range of 2030–2150 cm$^{-1}$, which is much lower than that reported for efficient catalysts, including Ru/CaH$_2$. This can be clearly attributed to the high electron-donating capability of Ru/CaFH at room temperature. H$_2$-TPD (Fig. 2a) and DFT experiments revealed that the abstraction of H atoms from CaFH by Ru forms CaFH with H$^-$ defects trapping electrons at ca. 50 °C, resulting in strong electron-donating power. Supplementary Fig. 10 shows an FT-IR spectrum (1550–1700 cm$^{-1}$) for N-adsorbed Ru/CaFH at 25 °C (N$_2$: 12 kPa) and a band assignable to $\delta$NH bending, which can be attributed to adsorbed ammonia from the broad band at ca.

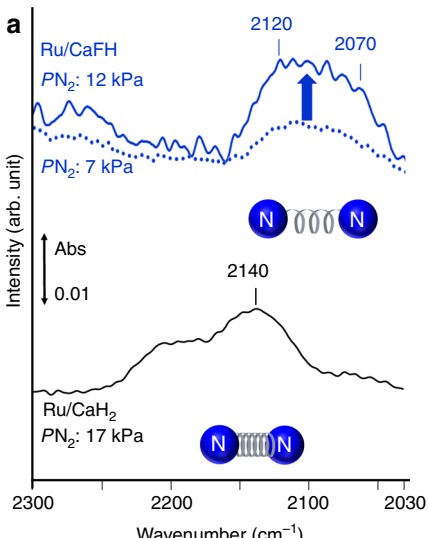

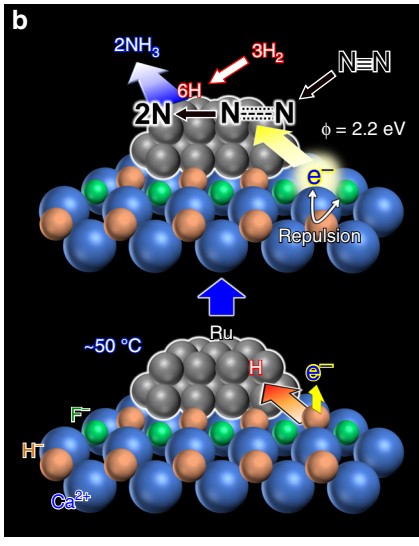

**Fig. 5 FT-IR spectra and a possible reaction mechanism. a** FT-IR spectra for $N_2$ adsorption on Ru/CaFH and Ru/CaH$_2$ after ammonia synthesis reaction at 340 °C, followed by cooling down below 20 °C. The FT-IR spectra were measured according to previous reports[19]. $N_2$ was adsorbed on the catalysts at 25 °C in the presence of $N_2$ (6–15 kPa). N≡N stretching ($\nu N_2$) bands were not observed in both catalysts under vacuum. **b** Proposed reaction mechanism.

1600 cm$^{-1}$. This indicates that $N_2$ molecules are dissociated into N adatoms, which react with H from CaFH even at 25 °C, and is consistent with the H$_2$-TPD results and ammonia synthesis from $N_2$ and D$_2$.

To summarize these results, Fig. 5b shows a proposed reaction mechanism. In Ru/CaFH where a (111) surface is probably formed in the CaFH solid solution due to its stability, the bond strength of Ca−F surpasses that of Ca−H, which weakens Ca−H bonds in the CaFH solid solution, so that Ru can abstract H atoms from H$^-$ sites in CaFH to leave electrons (e$^-$) in these sites, even at ca. 50 °C. The resultant CaFH with H$^-$ vacancies that trap e$^-$ behaves as a surface electride with a small work function ($\Phi = 2.2$ eV) that is much smaller than that of CaH$_2$ (2.7 eV) and comparable with that of metallic potassium (2.3 eV). This high electron-donating capability originates from the strong electron repulsion between e$^-$ in the H$^-$ and F$^-$ vacancies, which significantly enhances electron donation from the electride to the $\pi^*$ orbitals of $N_2$ molecules through the 3d orbitals of Ru. This facilitates the dissociative adsorption of $N_2$, which results in ammonia formation at low temperatures.

## Methods

**Preparation of CaH$_2$–BaF$_2$ mixture and Ru/CaFH**. First, a mixture of 87 mol% BaF$_2$ (Wako Chemicals) and 13 mol% BaH$_2$ (Stream Chemicals) was heated at 400 °C for 10 h in a flow of Ar (15 mL min$^{-1}$) to put large and rigid BaF$_2$ particles into disorder. The XRD pattern of the resulting material (modified BaF$_2$) consisted of broad diffraction peaks due to BaF$_2$ (Supplementary Fig. 11). No diffraction peaks due to BaH$_2$ were observed in the XRD pattern of modified BaF$_2$. Next, a simple mixture of CaH$_2$ (Sigma-Aldrich) and the resulting modified BaF$_2$ powders (Ca/Ba atomic ratios of 98:2, 9:1, 5:1, and 3:1) was heated at 340 °C for 10 h in a flow of H$_2$ (2.5 mL min$^{-1}$).

Ru/CaFH was prepared by two methods based on chemical vapor deposition using ruthenium acetylacetonate (Ru(acac)$_3$). In Method 1, the resulting material (Ca/Ba atomic ratio of 98:2) obtained by the above procedure was mixed with Ru(acac)$_3$ (Sigma-Aldrich) for deposition of 12 wt% Ru at 260 °C for 2 h and at 340 °C for 10 h in a flow of H$_2$ (2.5 mL min$^{-1}$). In Method 2, Ru nanoparticles were loaded onto a mixture of CaH$_2$ and BaF$_2$ by chemical vapor deposition using Ru(acac)$_3$ for deposition of 12 wt% Ru. The mixture of 98 mol% CaH$_2$, 2 mol% modified BaF$_2$, and Ru(acac)$_3$ corresponding to 12 wt% Ru was heated at 260 °C for 2 h and at 340 °C for 10 h in a flow of H$_2$ (2.5 mL min$^{-1}$). There was no significant difference in ammonia synthetic activity and structure between the catalysts prepared by both methods. The results for the catalyst prepared by simple Method 2 are shown in this paper.

**Preparation of solid solution CaF$_x$H$_{2-x}$–CaF$_2$ and Ru/CaF$_x$H$_{2-x}$–CaF$_2$**. Appropriate amounts of CaF$_2$ (Wako Chemicals) and CaH$_2$ were mixed by grinding, and the mixture was heated at 550 °C for 20 h in a flow of Ar (15 mL min$^{-1}$), which resulted in CaF$_x$H$_{2-x}$ solid solutions ($1.0 \leq x \leq 1.6$). Ru/CaF$_x$H$_{2-x}$–CaF$_2$ ($x = 1$) was prepared by heating CaF$_x$H$_{2-x}$–CaF$_2$ ($x = 1$) and Ru(acac)$_3$ corresponding to 12 wt% Ru at 260 °C in a flow of H$_2$ (2.5 mL min$^{-1}$). After 2 h, the sample was heated at 340 °C for 10 h in the H$_2$ flow.

**Preparation of Ru/CaH$_2$, Ru/BaH$_2$, and Ru/BaH$_2$–BaO**. According to previous reports[18], Ru/CaH$_2$, Ru/BaH$_2$, and Ru/BaH$_2$–BaO were prepared by heating CaH$_2$, BaH$_2$, and a mixture of 3 mol% BaO (Kojundo Chemical) and 97 mol% CaH$_2$, respectively, with Ru(acac)$_3$ corresponding to 10 wt% Ru at 260 °C in a flow of H$_2$ (2.5 mL min$^{-1}$). After 2 h, the samples were heated at 340 °C for 10 h in the H$_2$ flow. The activities of both catalysts for ammonia synthesis increased with the Ru loading and reached their respective maximum at a loading of 10 wt% Ru.

**Preparation of Cs–Ru/MgO**. Cs–Ru/MgO was prepared according to previous reports[2]. MgO (Ube, 500A) was heated in high vacuum at 500 °C for 6 h and then stirred in a solution of Ru$_3$(CO)$_{12}$ in THF for 4 h at room temperature. After evaporating the solvent, the obtained powder was slowly heated to 450 °C in high vacuum to decompose the carbonyl precursor. The amount of Ru loading was 10 wt%. After the obtained gray powder was stirred in a solution of Cs$_2$CO$_3$ in dehydrated ethanol for 3 h, the solvent was evaporated. The resulting catalyst was dried in vacuum. The catalytic activity of Cs–Ru/MgO reached a maximum at loading of 10 wt% Ru.

**Evaluation of catalytic performance**. Typical ammonia synthesis was conducted in a silica-glass fixed-bed reactor (catalyst: 0.1 g) in a flow of $N_2 − H_2$ ($N_2$:H$_2$ = 1:3, 60 mL min$^{-1}$, weight hourly space velocity (WHSV): 36,000 mL gcat$^{-1}$ h$^{-1}$) under atmospheric pressure (0.1 MPa). First, ammonia synthesis over each tested catalyst was conducted at 340 °C under the specified reaction conditions. When no increase or decrease in activity was observed for over 30 h, the catalyst was cooled down below 20 °C in a flow of $N_2$ at a flow rate of 60 mL min$^{-1}$ and then held under this flow for 5 h. After no ammonia was detected, the catalyst was heated at specific temperatures in a flow of $N_2$–H$_2$ ($N_2$:H$_2$ = 5:1, 60 mL min$^{-1}$). Ammonia was analyzed by both direct mass spectrometry (BELMass, MicrotracBEL, Japan) and ion chromatography. There was no difference in ammonia formation rate between both methods. In the case of ion chromatography, the ammonia produced was trapped in 5 mM H$_2$SO$_4$ aqueous solution, and the amount of NH$_4^+$ generated in the solution was estimated using an ion chromatograph (LC-2000 plus, Jasco) equipped with a conductivity detector. The rate of ammonia formation was repeatedly measured more than three times after the ammonia formation rate remained constant for over 1 h. It was verified that the measured rate had an error of less than 10%. There was no difference in ammonia formation rate between direct mass spectrometry and ion chromatography.

**Ammonia synthesis from $N_2$ and $D_2$.** A mixture of 98 mol% $CaH_2$, 2 mol% modified $BaF_2$, and $Ru(acac)_3$ corresponding to 12 wt% Ru was heated in the reactor at 260 °C for 2 h and at 300 °C for 10 h in a flow of $H_2$ (2.5 mL min$^{-1}$). The resultant Ru/CaFH was cooled down from that temperature to 180 °C in a flow of Ar at a flow rate of 2.5 mL min$^{-1}$ and held under this flow for 5 h. After each mass signal intensity was kept constant, $N_2–D_2$ ($N_2$: 5 mL min$^{-1}$, $D_2$: 15 mL min$^{-1}$) was passed into Ru/CaFH at that temperature under atmospheric pressure. The outlet gas from the reactor was analyzed using mass spectrometry (BELMass, MicrotracBEL, Japan).

**Characterization.** XRD (D8 Advance, Bruker) patterns were obtained using Cu Kα radiation. Nitrogen adsorption–desorption isotherms were measured at –196 °C with a surface-area analyzer (BELSORP-mini II, MicrotracBEL) to estimate the Brunauer–Emmett–Teller (BET) surface areas. The morphology of the samples was examined using scanning electron microscopy (SEM, S-5500, Hitachi) equipped with an energy-dispersive X-ray spectroscopy (EDX, EMAX EX-250, Horiba) detector. The microstructural characteristics of the samples were determined from transmission electron microscopy (TEM, JEM-ARM 200 F, Jeol) observations. $H_2$-TPD measurements were conducted by heating (1 °C min$^{-1}$) a sample (ca. 100 mg) in a stream of Ar (30 mL min$^{-1}$) and monitoring of the concentration of $H_2$ with a thermal conductivity detector (TCD) and a mass spectrometer (BELMass, MicrotracBEL, Japan). FT-IR spectroscopy measurements of adsorbed $N_2$ were conducted using a spectrometer (FT/IR-6100, Jasco) equipped with a mercury–cadmium–tellurium detector at a resolution of 4 cm$^{-1}$. Samples were pressed into self-supported disks. A disk was placed in a sealed and Ar-filled silica-glass cell equipped with NaCl windows to a closed gas-circulation system to allow thermal adsorption–desorption experiments. The disk was heated under vacuum at 200 °C for 90 min. After the pretreatment, the disk was cooled to 25 °C under vacuum to obtain a background spectrum from the spectra of the $N_2$-adsorbed samples. Pure $N_2$ (99.99995%) was supplied to the system through a liquid-nitrogen trap. In all, 12 wt% Ru/$CaH_2$ and 12 wt% Ru/CaFH were used for FT-IR measurements. XPS (ESCA-3200, Shimadzu, Mg Kα, 8 kV, 25 mA) was performed in conjunction with an Ar-filled glovebox. The samples were moved to the ultra-high-vacuum (UHV) XPS apparatus through the Ar-filled glovebox without exposure to the ambient air. The binding energy was corrected with respect to the Au 4f$_{7/2}$ peak of Au-deposited samples.

**DFT computations.** The work functions of near-surface ions were calculated using the slab supercell model on the (111) plane, which is the most experimentally (cleavage plane) and theoretically (smallest surface energy) stable plane for fluorite crystals. The slabs were constructed by relaxing a bulk unit cell of cubic CaFH (5.465 A)[17] and then stacking up the relaxed cell to form a slab. The slab was repeated periodically in the [111] direction with a vacuum gap of >20 Å to form a supercell. Surface relaxation was taken into account by further relaxation of the two layers on each surface of the slabs. Relaxation was terminated when the force on each atom became less than 0.01 eV Å$^{-1}$. All the slab calculations were performed with a cut-off energy of 500 eV and a k mesh of $2 \times 2 \times 1$. The slab thickness was varied between seven and ten layers to ascertain convergence with respect to thickness. The band structures for CaFH(111) solid solutions with H$^-$ or F$^-$ defects (CaFH$_{1-x}$, CaF$_{1-x}$H) were computed in this study. The results are shown in Supplementary Fig. 5. The work function of CaFH$_{1-x}$ is identical to that of CaF$_{1-x}$H, and was estimated to be 2.2 eV. This value is smaller than that for $CaH_2$ (2.7 eV), which suggests that fluorine with a larger electronegativity pushes the energy level of electrons up by electrostatic repulsion.

## Data availability

All relevant data that support the findings of this study are presented in the paper and supporting information. Source data are available from the corresponding author upon reasonable request.

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

## Acknowledgements

This work was supported by a fund from the Grants-in-Aid for Japan Society for the Promotion of Science (JSPS) Fellows and for Scientific Research from the Ministry of Education, Culture, Science, Sports, and Technology (MEXT) of Japan (19K15358, 18H05251, and 17H06153).

## Author contributions

M. Hattori designed the study, carried out catalyst synthesis, its evaluation, and mechanistic study, and composed the paper. S.I. performed catalyst evaluation and characterization. T.N. performed DFT computation. M. Hara and H.H. supervised the study, planned the experiments, and analyzed the data.

## Competing interests

The authors declare no competing interests.
