## [Peer Review File · Nature Communications]

Reviewers' Comments:

Reviewer #1:

Remarks to the Author:

Ammonia synthesis under low temperatures and pressures is an exciting research topic in chemistry. Hattori et al. reports an active Ru/CaFH catalyst which exhibits catalytic performance in ammonia synthesis at a temperature as low as 50 °C. This is not attainable on most of the previous reported Ru-based catalysts. The superior catalytic performance has been attributed to the high electron-donating capability of CaFH support. Replacing part of H in CaH₂ with F- lowers the temperature for formation of H- vacancies and increase the energy of electrons trapped at H- vacancies. Although the strategy and the performance of catalyst are impressive, however, this work has some big weaknesses and thus should not be published on Nature Communications.

1. The activity data collected below 200 °C should be carefully verified. As shown in the method section, the catalysts were pretreated in a N₂-H₂ gas at 340 °C, followed by cooling down 20 °C in a flow of N₂, He or H₂. Some N species would be expected present on/in the catalyst. Did they measure the N content in the sample after pretreatment? Although the authors mentioned that the ammonia was not observed under a flow of H₂ during the cooling procedure, this control experiment is not satisfactory. I will suggest they run the catalytic test at different temperatures without any pretreatment in a N₂-H₂ flow. Moreover, did they observe any ammonia during heating the pretreated sample in a pure H₂ flow with the same manner for testing?
2. In Table 1, were the activity data obtained under steady state?
3. It is quite strange that the support CaFH was prepared with CaH₂ and BaF₂. Why not CaH₂ and CaF₂? What is the activity of Ru/CaFH for which the CaFH was made from a mixture of CaH₂ and CaF₂? It is well known that Ba is an excellent promoter for Ru-based catalysts. Is it possible that Ba plays an important role in ammonia synthesis? This should be clarified because it is essentially relevant to the proposed reaction mechanism.
4. The apparent activation barrier (20 kJ/mol) derived from the Arrhenius plot, with only three points is highly questionable. In general, such a low apparent activation energy suggests the experiments might be measured at transport-limited regimes.
5. Did they have any information on the chemical state of Ru? Could they provide more information on the electron transfer from CaFH support to Ru metal?

Reviewer #2:

Remarks to the Author:

The authors report a catalyst for ammonia synthesis that displays activity at much lower operating temperatures (50 C) than conventional ammonia catalysts. The activation barrier for ammonia formation is lowered due to a CaFH phase that acts as a strong electron donor to the Ru catalyst. This study will be important for the catalysis community and provides a deeper understanding of catalytic processes. I therefore support the acceptance of this paper with modest revisions.

Specific comments:

- 1) Why did you not just use the pure CaFH phase for ammonia synthesis? Is there a reason that the CaF₂-BaF₂ (98:2 ratio) material performed better? This needs to be explained. If your current hypothesis is that the workfunction of Ru is lowered by CaFH because of its ability to donate electrons readily, pure CaFH should be a great catalyst.
- 2) More data is needed to claim that CaFH exists on the surface of the CaF₂-BaF₂ (98:2 ratio)

material. Your XRD does not show a peak for CaFH, which might be resolved with longer scan times. Also, your XPS and EDS results only show that there is fluorine on the surface, not that the signal is due to CaFH. Please provide your XPS data in the SI and provide further evidence that CaFH is present on the surface.

3) In line 199 of the main text, you claim that the Ru/CaFH catalyst surpasses conventional catalysts that operate at high temperature. Do you mean this in terms of operating temperature or rate of ammonia formation? In Supplementary Table 1, the catalysts listed have much larger rates of ammonia formation ($> 1000 \text{ umol/g*hr}$) than your listed values in Fig. 1 ($\sim 100 \text{ umol/g*hr}$ at 100 C). What was the ammonia formation rate of Ru/CaFH at higher temperatures? I would suggest adding this information to Supplementary Table 1.

4) In Fig. 3, please label the onset of H₂ desorption for the CaFH material. At first glance, it is not apparent that there is a difference with CaH₂

5) In line 132 of the main text, you claim that the XRD peaks for BaH₂ can be observed in Fig. 4a. This is not apparent. I would suggest zooming in on the BaH₂ peaks just as you have done for the CaFH peaks or improve the resolution of your diffractogram.

6) The authors have provided limited evidence for formation of electride-type electrons on the surface of CaFH. I agree that this is a plausible hypothesis. However, in absence of better support, the authors should re-write their abstract, as it overstates the evidence. Specifically, the authors wrote: "the high electron-donating capability due to the low work function (2.2 eV) of H⁻ vacancies that trap electrons." This is stated with too much certainty.

Reviewer #3:

Remarks to the Author:
see the review attached

The authors applied a solid mixture of CaF_2 and CaH_2 in ammonia synthesis, which showed small activity at 50°C with a low activation energy barrier of 20 kJ mol^{-1} . But, there are many issues yet remained to answer than it can offer. In its present state, the manuscript is not considered suitable for publication in Nature Communications for the following reasons:

1. It is certainly 'eye-catching' and attractive to develop ammonia synthesis as low as 50°C at ambient temperature, instead of using conventional $400\text{-}450^\circ\text{C}$ of over 150 bars. However, there have been numerous issues that cannot be achieved such goal in the past.

First, the authors used thermodynamics equilibrium and heat of reactions to justify the needs for low temperature development of this reaction using the wind energy. However, from catalysis point of view, their argument does not carry any useful sense to lead to the catalysis development.

For all exothermic catalytic reactions including ammonia synthesis, if one can achieve equilibrium, there is of course to argue that the equilibrium will become unfavourable at higher temperatures, hence using low temperatures would be an advantage.

But, the problem is that whether one can overcome the kinetic barrier at low temperature to achieve the equilibrium. Many industrial processes are exothermic but still be carried out at elevated temperatures due to unfavourable low rate at low temperature rather than their equilibrium position. If the conventional catalysts can get higher rates for ammonia synthesis at lower temperature, then the present extreme conditions would not be used: whether this is for non-renewables or wind energy, it is not the key factor. Using pressure not only affect the equilibrium position but also increase the kinetics.

I have found that their rate of ammonia production at low temperature is very poor (many folds below the conventional catalysts at elevated temperature). Can they challenge the conventional catalysts? I really doubt it. Any defects/ impurities to create excited state in the catalyst can give miniscule activity at low temperatures but can they obtain an acceptable yields without much recycling ammonia in technical feasibility way, there is no evidence on this paper. At elevated temperatures whether CaFH survives and still dominants the activity (they seemed to have made this assumption) over other higher activated sites? There is also no discussion.

2. The relationship between the Ru nanoparticles and ionic support is not discussed. The author used the bulk mechanism to argue the formation of H donor sites and electron donor sites (low work function). The DFT calculations and XRD studies to imply their thermodynamic trend to contribute to Ru is not useful. But, how can these ionic centers move from bulk ionic compound to the Ru surface? If only the surface contact, then some surface studies are required.
3. H_2 -TPD was the only experimental evidence to prove the electron trapping at low temperature. However, the experiment was very roughly performed. The baseline was not even stabilised (Fig. 3 and Fig. S2). The poor signal to noise level cannot be used to claim the onset temperature of H_2 desorption.
4. More experiments for example in-situ EPR, XPS et al. are required to support the existence of vacancy and electron trapping.
5. Quality of SEM images is also poor for statistical counting of particle size.

6. Catalyst weight required for equilibrium yield (CWEY) was employed to show the activity of catalysts: it is a crude and can easily mislead readers. Many other factors such as catalyst porosity, particle size and support can affect the activity when amplifying the catalysts amount.
7. The FT-IR spectra for N_2 adsorption was employed to show the back donation of electron from Ru to N_2 . As discussed, the H in the support CaFH can be used for hydrogenation of adsorbed N_2 . To illustrate their exchange of atomic/molecular species at both low and high temperature, the authors should label the NH species to demonstrate the interaction between the ionic phase and Ru.
8. HF may be formed under the reaction conditions, can the CaFH be stable at high temperature?
9. There are some format and spelling mistakes.

Reviewer #1 (Remarks to the Author):

Ammonia synthesis under low temperatures and pressures is an exciting research topic in chemistry. Hattori et al. reports an active Ru/CaFH catalyst which exhibits catalytic performance in ammonia synthesis at a temperature as low as 50 °C. This is not attainable on most of the previous reported Ru-based catalysts. The superior catalytic performance has been attributed to the high electron-donating capability of CaFH support. Replacing part of H in CaH₂ with F- lowers the temperature for formation of H- vacancies and increase the energy of electrons trapped at H- vacancies. Although the strategy and the performance of catalyst are impressive, however, this work has some big weaknesses and thus should not be published on Nature Communications.

1. The activity data collected below 200 °C should be carefully verified. As shown in the method section, the catalysts were pretreated in a N₂-H₂ gas at 340 °C, followed by cooling down 20 °C in a flow of N₂, He or H₂. Some N species would be expected present on/in the catalyst. Did they measure the N content in the sample after pretreatment? Although the authors mentioned that the ammonia was not observed under a flow of H₂ during the cooling procedure, this control experiment is not satisfactory. I will suggest they run the catalytic test at different temperatures without any pretreatment in a N₂-H₂ flow. Moreover, did they observe any ammonia during heating the pretreated sample in a pure H₂ flow with the same manner for testing?

We accept the reviewer's criticism concerning surface N species formed on Ru/CaFH by preparation using N₂-H₂ gas at 340 °C. According to the reviewer's suggestion, the catalyst was prepared in a flow of pure H₂ gas alone for each measurement of the ammonia formation. It was confirmed by X-ray photoelectron spectroscopy (XPS) measurements that the prepared Ru/CaFH samples had no surface N species. The reduction of deposited Ru species into metallic Ru requires heating at ≥ 300 °C in the presence of H₂. There was no difference in activity and stability at each temperature (50-340 °C) between the catalysts pretreated with only H₂ and with the N₂-H₂ mixture; therefore, no N species are formed by activation in a flow of N₂-H₂ at 340 °C and there is no contribution from surface N species to low temperature ammonia synthesis over Ru/CaFH.

We suppose that the role of alkaline earth metal nitrides in Ru catalytic systems, where Ru has a strong affinity for N, is distinct from those in other transition metal catalytic systems.

According to reviewer's suggestion, we try catalytic test at different temperature with the suggested manner. There is no significant difference in ammonia formation rate between the sample pretreated in N₂-H₂ and H₂.

The following has been added to "**1.2. N species formed on catalysts by activation at 340 °C before low temperature ammonia synthesis**" in **1. Additional discussion** of Supplementary Information in the revised manuscript.

Page 2, line 12-17.

"In this study, low temperature ammonia synthesis was carried out through ammonia synthesis in a flow of N₂-H₂ at 340 °C, followed by cooling down below 20 °C in a flow of N₂. It was confirmed by XPS measurements that the prepared Ru/CaFH samples had no surface N species. Moreover, when the tested catalysts, including Ru/CaFH, cooled down below 20 °C were heated in a flow of H₂ or He at 1 °C min⁻¹, the desorption of ammonia and N₂ was not detected at all even by a mass spectrometer."

2. In Table 1, were the activity data obtained under steady state?

The rates of ammonia formation were measured under steady-state, as pointed out by the reviewer.

The following has been added to “**Methods**” in the revised manuscript.

Page 11, line 25-27.

“The rate of ammonia formation was repeatedly measured more than 3 times after the ammonia formation rate remained constant for over 1 h.”

3. It is quite strange that the support CaFH was prepared with CaH₂ and BaF₂. Why not CaH₂ and CaF₂? What is the activity of Ru/CaFH for which the CaFH was made from a mixture of CaH₂ and CaF₂? It is well known that Ba is an excellent promoter for Ru-based catalysts. Is it possible that Ba plays an important role in ammonia synthesis? This should be clarified because it is essentially relevant to the proposed reaction mechanism.

We appreciate the reviewer’s comment, and this is one of the highlights in this study, although we did not emphasize it in the manuscript. The surface areas of the supports and catalytic activities of Ru/CaFH, Ru/CaF_xH_{2-x}-CaF₂ (x=1) and Ru/BaH₂ at 340 °C have been added to the revised manuscript as Supplementary Table 4. Ru/CaF_xH_{2-x}-CaF₂ (x=1) prepared from CaH₂ and CaF₂ may have potential to act as an effective electron-donating material. However, the surface area of the material obtained by conventional solid-state reaction at the required high heating temperature (≥550 °C) for 20 h is so small (1 m² g⁻¹) that it cannot exhibit sufficient catalytic performance. In this study, the addition of a small amount of BaF₂ to CaH₂ was found to form a CaFH solid solution on CaH₂ at lower temperatures, which resulted in a surface area of 10 m² g⁻¹. The catalytic performance of Ru/CaFH can be partly attributed to the surface area achieved with the new method.

For the Ru/CaFH system, the added BaF₂ is converted into BaH₂ with the formation of the CaF_{1.0}H_{1.0} solid solution, as shown in Fig. 2b, and Ru nanoparticles with deposited BaH₂ (Ru/BaH₂, Supplementary Table 4 in the revised manuscript) showed a much smaller catalytic activity than Ru/CaF_xH_{2-x}-CaF₂ (x=1); therefore, the catalytic performance of Ru/CaFH can be attributed to the CaF_{1.0}H_{1.0} solid solution, rather than to Ba species.

The surface areas of the supports and catalytic activities of Ru/CaFH, Ru/CaF_xH_{2-x}-CaF₂ (x=1) and Ru/BaH₂ at 340 °C have been added to the revised manuscript as Supplementary Table 4. Furthermore, the following has been added to “**Catalytic performance for ammonia synthesis at low temperatures**” in the revised manuscript.

Page 7, line 19-35.

“The surface areas of the supports and ammonia formation rates (340 °C) of Ru/CaFH, Ru-deposited BaH₂ (Ru/BaH₂) and Ru/CaF_xH_{2-x}-CaF₂ (x=1) where Ru nanoparticles are deposited on the CaF_{1.0}H_{1.0} solid solution formed by heating mixtures of orthorhombic CaH₂ and cubic CaF₂ at 550 °C are summarized in Supplementary Table 4. BaH₂ and CaF_{1.0}H_{1.0} solid solution are expected to be formed on Ru/CaFH, and either or both of them can contribute to the catalytic performance of Ru/CaFH. However, Ru/BaH₂ had a much smaller catalytic activity for ammonia synthesis than Ru/CaF_xH_{2-x}-CaF₂ (x=1), which indicates that the catalysis of Ru/CaFH is derived from the CaF_{1.0}H_{1.0} solid solution. Supplementary Table 4 also shows Ru/CaF_xH_{2-x}-CaF₂ (x=1) to be inferior to Ru/CaFH with respect to ammonia synthesis. This can be attributed to the conventional preparation method for the CaF_{1.0}H_{1.0} solid solution. Ru/CaF_xH_{2-x}-CaF₂ (x=1) was prepared by the deposition of Ru nanoparticles. CaF_xH_{2-x}-CaF₂ (x=1) synthesized by high temperature solid-state reaction at ≥550 °C for 20 h and the resultant surface area was very small (1 m² g⁻¹), which limits the catalytic activity of Ru/CaF_xH_{2-x}-CaF₂ (x=1). On the other hand, in the case of CaFH prepared by the new method, CaFH solid solution

was formed on the surface with a surface area of $10 \text{ m}^2 \text{ g}^{-1}$. This is considered to be the reason for the difference in activity between Ru/CaFH and Ru/CaF_xH_{2-x}-CaF₂ (x=1).”

Furthermore, “**Preparation of Ru/CaH₂ and Ru/BaH₂-BaO**” in “**Methods**” (Page 10, line 30-Page 11, line 1.) has been replaced by the following in the revised manuscript.

“Preparation of Ru/CaH₂, Ru/BaH₂ and Ru/BaH₂-BaO. According to previous reports¹, Ru/CaH₂, Ru/BaH₂ and Ru/BaH₂-BaO were prepared by heating CaH₂, BaH₂ and a mixture of 3 mol% BaO (Kojundo Chemical) and 97 mol% CaH₂, respectively, with Ru(acac)₃ corresponding to 10 wt% Ru at 260 °C in a flow of H₂ (2.5 mL min⁻¹). After 2 h, the samples were heated at 340 °C for 10 h in the H₂ flow. The activities of both catalysts for ammonia synthetic increased with the Ru loading and reached their respective maximum at a loading of 10 wt% Ru.”

4. The apparent activation barrier (20 kJ/mol) derived from the Arrhenius plot, with only three points is highly questionable. In general, such a low apparent activation energy suggests the experiments might be measured at transport-limited regimes.

As pointed out by the reviewer, the explanation for Table 1 was not clear. The E_a in Table 1 was estimated from the ammonia formation rates at 50, 75, 100 and 125 °C although this was described as 50-150 °C written in the original manuscript. The coefficient of determination (r²) for the Arrhenius equation obtained from the rates was 0.992. The temperature range “50-150 °C” has been replaced by “50-125 °C” in Table 1, and the ammonia formation rates for Ru/CaFH at 50, 75, 100 and 125 °C (50, 75, 120 and 190 μmol g⁻¹ h⁻¹) have been added to the footnote of Table 1 in the revised manuscript.

The followings imply that the experiments are not in the mass transport regime.

1. The rate of ammonia formation over Ru/RuCaFH was measured at various catalyst amounts and flow rates, but at the same WHSV (36000 mL gcat⁻¹ h⁻¹), and it was confirmed that the same WHSV gives the same ammonia formation rate. This indicates that external diffusion limitation can be eliminated from the reaction system. Nitrogen adsorption–desorption isotherm and Barrett–Joyner–Halenda (BJH) pore-size distribution revealed that Ru/CaFH has no small pore structures (Supplementary Table 2 in the revised manuscript), causing internal diffusion limitation.
2. The apparent activation energy for ammonia formation over Cs-Ru/MgO, a benchmark catalyst, was estimated at 94 kJ mol⁻¹ (200, 250, 300 and 340 °C) under our reaction conditions. This was the same as those of Cs-Ru/MgO reported by other groups (for example, Reference 21 in the main text).
3. In general, reaction at higher temperatures is subject to mass transport regime.

5. Did they have any information on the chemical state of Ru? Could they provide more information on the electron transfer from CaFH support to Ru metal?

According to the reviewer’s comment, the similarity of XPS Ru 3p_{3/2} spectra for Ru/CaFH and Ru-deposited SiO₂ (the average Ru particle size (3.2 nm) to that for Ru/CaFH (3.4 nm)) has been shown as Supplementary Fig. 9 in the revised manuscript. Ru 3p_{3/2} for Ru/CaFH appeared at a slightly lower binding energy than that of Ru/SiO₂, which indicates that Ru on CaFH is more negative than Ru on SiO₂. However, the difference is so small that we cannot evaluate the electron donation from CaFH to Ru. This is because XPS reflects only Ru atoms near the edges of Ru particles connected to CaFH from the point of view of the escape depth of photoelectrons. For this reason, we have adopted Fourier transform infrared (FT-IR) spectroscopy measurements using N₂ as a probe molecule, which is a more sensitive method. νN₂ for N₂ adsorbed on SiO₂

was observed at $>2200\text{ cm}^{-1}$, whereas that for Ru/CaFH appears below 2150 cm^{-1} .

The following has been added to “**Electron-donating capability and reaction mechanism for Ru/CaFH**” in the revised manuscript.

Page 8, line 19-26.

“XPS Ru $3p_{3/2}$ of Ru/CaFH was used to evaluate the electron-donating capability from CaFH to Ru (Supplementary Fig. 9). Ru $3p_{3/2}$ for Ru/CaFH appeared at a slightly lower binding energy than that for metallic Ru particles deposited on SiO_2 (Ru/ SiO_2), which indicates that Ru on CaFH is more negative than Ru on SiO_2 . However, the difference was not so large, because XPS reflects only Ru atoms near the edges of Ru particles connected to CaFH from the point of view of the escape depth of photoelectrons. For this reason, we have adopted Fourier transform infrared (FT-IR) spectroscopy measurements using N_2 as a probe molecule, which is a more sensitive method (FT-IR, Fig. 4).”

Reviewer #2 (Remarks to the Author):

The authors report a catalyst for ammonia synthesis that displays activity at much lower operating temperatures (50 C) than conventional ammonia catalysts. The activation barrier for ammonia formation is lowered due to a CaFH phase that acts as a strong electron donor to the Ru catalyst. This study will be important for the catalysis community and provides a deeper understanding of catalytic processes. I therefore support the acceptance of this paper with modest revisions.

Specific comments:

1) Why did you not just use the pure CaFH phase for ammonia synthesis? Is there a reason that the $\text{CaF}_2\text{-BaF}_2$ (98:2 ratio) material performed better? This needs to be explained. If your current hypothesis is that the workfunction of Ru is lowered by CaFH because of its ability to donate electrons readily, pure CaFH should be a great catalyst.

We appreciate the reviewer’s comment, and this is one of the highlights of this study. The surface areas of the supports and catalytic activities of Ru/CaFH, Ru/ $\text{CaF}_x\text{H}_{2-x}\text{-CaF}_2$ ($x=1$) and Ru/ BaH_2 at $340\text{ }^\circ\text{C}$ have been added to the revised manuscript as Supplementary Table 4. Ru/ $\text{CaF}_x\text{H}_{2-x}\text{-CaF}_2$ ($x=1$) prepared from CaH_2 and CaF_2 may have the potential to act as an effective electron-donating material. However, the surface area of the material obtained by conventional solid-state reaction at the required high heating temperature ($\geq 550\text{ }^\circ\text{C}$) for 20 h is so small ($1\text{ m}^2\text{ g}^{-1}$) that it cannot exhibit sufficient catalytic performance. In this study, the addition of a small amount of BaF_2 to CaH_2 was found to form a CaFH solid solution on CaH_2 at lower temperatures, which resulted in a surface area of $10\text{ m}^2\text{ g}^{-1}$. The catalytic performance of Ru/CaFH can be partly attributed to the surface area achieved with the new method.

The surface areas of the supports and catalytic activities of Ru/CaFH, Ru/ $\text{CaF}_x\text{H}_{2-x}\text{-CaF}_2$ ($x=1$) and Ru/ BaH_2 at $340\text{ }^\circ\text{C}$ have been added to the revised manuscript as Supplementary Table 4. Furthermore, the following has been added to “**Catalytic performance for ammonia synthesis at low temperatures**” in the revised manuscript.

Page 7, line 19-35.

“The surface areas of the supports and ammonia formation rates ($340\text{ }^\circ\text{C}$) of Ru/CaFH, Ru-deposited BaH_2 (Ru/ BaH_2) and Ru/ $\text{CaF}_x\text{H}_{2-x}\text{-CaF}_2$ ($x=1$) where Ru nanoparticles are deposited on the $\text{CaF}_{1.0}\text{H}_{1.0}$ solid solution formed by heating mixtures of orthorhombic CaH_2

and cubic CaF_2 at 550 °C are summarized in Supplementary Table 4. BaH_2 and $\text{CaF}_{1.0}\text{H}_{1.0}$ solid solution are expected to be formed on Ru/CaFH, and either or both of them can contribute to the catalytic performance of Ru/CaFH. However, Ru/ BaH_2 had a much smaller catalytic activity for ammonia synthesis than Ru/ $\text{CaF}_x\text{H}_{2-x}\text{-CaF}_2$ ($x=1$), which indicates that the catalysis of Ru/CaFH is derived from the $\text{CaF}_{1.0}\text{H}_{1.0}$ solid solution. Supplementary Table 4 also shows Ru/ $\text{CaF}_x\text{H}_{2-x}\text{-CaF}_2$ ($x=1$) to be inferior to Ru/CaFH with respect to ammonia synthesis. This can be attributed to the conventional preparation method for the $\text{CaF}_{1.0}\text{H}_{1.0}$ solid solution. Ru/ $\text{CaF}_x\text{H}_{2-x}\text{-CaF}_2$ ($x=1$) was prepared by the deposition of Ru nanoparticles. $\text{CaF}_x\text{H}_{2-x}\text{-CaF}_2$ ($x=1$) synthesized by high temperature solid-state reaction at ≥ 550 °C for 20 h and the resultant surface area was very small ($1 \text{ m}^2 \text{ g}^{-1}$), which limits the catalytic activity of Ru/ $\text{CaF}_x\text{H}_{2-x}\text{-CaF}_2$ ($x=1$). On the other hand, in the case of CaFH prepared by the new method, CaFH solid solution was formed on the surface with a surface area of $10 \text{ m}^2 \text{ g}^{-1}$. This is considered to be the reason for the difference in activity between Ru/CaFH and Ru/ $\text{CaF}_x\text{H}_{2-x}\text{-CaF}_2$ ($x=1$)."

2) More data is needed to claim that CaFH exists on the surface of the $\text{CaF}_2\text{-BaF}_2$ (98:2 ratio) material. Your XRD does not show a peak for CaFH, which might be resolved with longer scan times. Also, your XPS and EDS results only show that there is fluorine on the surface, not that the signal is due to CaFH. Please provide your XPS data in the SI and provide further evidence that CaFH is present on the surface.

We could obtain a clearer $\text{CaF}_{1.0}\text{H}_{1.0}$ (200) diffraction peak on the sample of Ca:Ba = 98:2, according to the reviewer's suggestion. The enlarged diffraction peak has been added to Fig. 2c in the revised manuscript.

We also measured Ca 2p XPS spectra for heated $\text{CaH}_2\text{-BaF}_2$ mixtures (Ca/Ba atomic ratio of 98:2), $\text{CaF}_x\text{H}_{2-x}\text{-CaF}_2$ ($x=1$) and CaH_2 according to the reviewer's suggestion and have added the results to the revised manuscript as Supplementary Fig. 3. The following description has been also added to the revised manuscript.

Page 5, line 12-17.

"In X-ray photoelectron spectroscopy (XPS) measurements for heated $\text{CaH}_2\text{-BaF}_2$ mixtures (Ca/Ba atomic ratio of 98:2), $\text{CaF}_x\text{H}_{2-x}\text{-CaF}_2$ ($x=1$) and CaH_2 (Supplementary Fig. 3), the Ca 2p peaks for both heated $\text{CaH}_2\text{-BaF}_2$ mixtures and $\text{CaF}_x\text{H}_{2-x}\text{-CaF}_2$ ($x=1$) appeared at 346.5 eV, which is lower than that for CaH_2 (347.3 eV) and supports the formation of the $\text{CaF}_{1.0}\text{H}_{1.0}$ solid solution on the surface of a small amount of BaF_2 -added CaH_2 ."

3) In line 199 of the main text, you claim that the Ru/CaFH catalyst surpasses conventional catalysts that operate at high temperature. Do you mean this in terms of operating temperature or rate of ammonia formation? In Supplementary Table 1, the catalysts listed have much larger rates of ammonia formation ($> 1000 \text{ umol/g*hr}$) than your listed values in Fig. 1 ($\sim 100 \text{ umol/g*hr}$ at 100 C). What was the ammonia formation rate of Ru/CaFH at higher temperatures? I would suggest adding this information to Supplementary Table 1.

I would like to express our gratitude to the reviewer for this comment. We have added the physicochemical information (surface area, porosity and Ru particle size) and the rates of ammonia formation for Ru/CaFH, Ru/ CaH_2 (comparison catalyst), Ru/ BaO-BaH_2 , Ru/ $\text{Ba-Ca}(\text{NH}_2)_2$, Ru/C12A7, and Cs-Ru/MgO and a commercial Fe catalyst (benchmark catalysts) at 100-340 °C to the revised manuscript as Supplementary Table 2, taking care to make the table easy to read.

The following with a reference paper (*Appl. Catal. General*, **142**, 209-222 (1996), Reference 20) has been added to the revised manuscript accordingly.

Page 6, line 19-25.

“Supplementary Table 2 shows the physicochemical information (surface area, porosity, and Ru particle size) and the rates of ammonia formation for Ru/CaFH, Ru/CaH₂, Ru/Ba-Ca(NH₂)₂, Ru/BaO-BaH₂, Ru/C12A7¹⁹, Cs-Ru/MgO and a commercial Fe catalyst as benchmark catalysts at 100-340 °C. It was confirmed that the catalytic activities of the commercial Fe catalyst and Cs-Ru/MgO benchmark catalysts were comparable to those reported by other groups¹⁹⁻²¹. These conventional catalysts did not exhibit activity for ammonia synthesis below 100-200 °C,”

4) In Fig. 3, please label the onset of H₂ desorption for the CaFH material. At first glance, it is not apparent that there is a difference with CaH₂.

Each H₂-TPD profile with the background profile was measured again and has been added to Fig. 3a and Supplementary Fig. 4 in the revised manuscript. According to the reviewer's suggestion, H₂ desorption from CaFH has been labeled in Fig. 2a of the revised manuscript. No desorption of molecular species giving the $m/z=2$ signal, such as H₂O, was observed.

5) In line 132 of the main text, you claim that the XRD peaks for BaH₂ can be observed in Fig. 4a. This is not apparent. I would suggest zooming in on the BaH₂ peaks just as you have done for the CaFH peaks or improve the resolution of your diffractogram.

We appreciate the reviewer's suggestion. The diffraction due to BaH₂ in CaFH was measured again with high resolution and has been enlarged in Fig. 2b of the revised manuscript.

6) The authors have provided limited evidence for formation of electride-type electrons on the surface of CaFH. I agree that this is a plausible hypothesis. However, in absence of better support, the authors should re-write their abstract, as it overstates the evidence. Specifically, the authors wrote: “the high electron-donating capability due to the low work function (2.2 eV) of H⁻ vacancies that trap electrons.” This is stated with too much certainty.

The reviewer's comment is justified. We accept the reviewer's criticism concerning the abstract and have deleted a portion in the revised manuscript.

Reviewer #3

The authors applied a solid mixture of CaF₂ and CaH₂ in ammonia synthesis, which showed small activity at 50 °C with a low activation energy barrier of 20 kJ mol⁻¹. But, there are many issues yet remained to answer than it can offer. In its present state, the manuscript is not considered suitable for publication in Nature Communications for the following reasons:

1. It is certainly ‘eye-catching’ and attractive to develop ammonia synthesis as low as 50 °C at ambient temperature, instead of using conventional 400-450 °C of over 150 bars. However, there have been numerous issues that cannot be achieved such goal in the past.

First, the authors used thermodynamics equilibrium and heat of reactions to justify the needs for low temperature development of this reaction using the wind energy. However, from catalysis point of view, their argument does not carry any useful sense to lead to the catalysis development.

For all exothermic catalytic reactions including ammonia synthesis, if one can achieve equilibrium, there is of course to argue that the equilibrium will become unfavorable at higher temperatures, hence using low temperatures would be an advantage.

But, the problem is that whether one can overcome the kinetic barrier at low temperature to achieve the equilibrium. Many industrial processes are exothermic but still be carried out at elevated temperatures due to unfavorable low rate at low temperature rather than their equilibrium position. If the conventional catalysts can get higher rates for ammonia synthesis at lower temperature, then the present extreme conditions would not be used: whether this is for non-renewables or wind energy, it is not the key factor. Using pressure not only affect the equilibrium position but also increase the kinetics.

I have found that their rate of ammonia production at low temperature is very poor (many folds below the conventional catalysts at elevated temperature). Can they challenge the conventional catalysts? I really doubt it. Any defects/impurities to create excited state in the catalyst can give miniscule activity at low temperatures but can they obtain an acceptable yield without much recycling ammonia in technical feasibility way, there is no evidence on this paper. At elevated temperatures whether CaFH survives and still dominates the activity (they seemed to have made this assumption) over other higher activated sites? There is also no discussion.

We should have paid attention to the problems pointed out by the reviewer and have provided sufficient explanation with appropriate results in the original manuscript. We accept the reviewer's criticism concerning the lack of explanation and discussion in the original manuscript, and have thus made corrections.

We have added the physicochemical information (surface area, porosity and Ru particle size) and the rates of ammonia formation for all tested catalysts (Ru/CaFH, Ru/CaH₂ (comparison catalyst), Ru/BaO-BaH₂, Ru/Ba-Ca(NH₂)₂, Ru/C12A7, Cs-Ru/MgO (benchmark catalyst) and commercial Fe catalyst (benchmark catalyst)) at 100-340 °C to the revised manuscript as Supplementary Table 2. The table clearly indicates that lowering the reaction temperature to 100-200 °C brings the rate of ammonia formation on all tested catalysts close to "zero". Ru/Ba-Ca(NH₂)₂ has been reported to have the highest catalytic performance for ammonia synthesis by our group (*Angew. Chem.* **130**, 2678 (2018))⁶; the ammonia formation rates are much higher than those of conventional catalysts reported as highly active catalysts, as shown in the literature and Supplementary Table 3 added to the revised manuscript. The catalytic performance of Ru/BaO-BaH₂ that was also reported by our group (*ACS Catal.* **8**, 10977 (2018))¹⁸ is close to Ru/Ba-Ca(NH₂)₂ (Supplementary Tables 2 and 3). It was confirmed that the catalytic activities of the two benchmark catalysts were comparable to those reported in the literature (see below). This implies that conventional catalysts equally lose activity for ammonia synthesis at 100-200 °C, even if they act as highly active catalysts above these temperatures. No catalyst that just manages to be active for ammonia synthesis from N₂ and H₂ below 100 °C has been reported.

As pointed out by the reviewer, more efficient ammonia synthesis from N₂ and H₂ is required to overcome the kinetic barrier at lower temperature to achieve the equilibrium. On the other hand, the present results suggest that the conventional approach, based on the expectation that catalysts with high activities at high reaction temperatures would also act as highly active catalysts at low temperatures, cannot perform at such lower reaction temperatures. It would be difficult to enhance the activity of catalyst at low temperatures without catalyst which can function for the reaction at the temperatures. In this study, we have adopted a new strategy to lower the temperature as the first step to achieve the desired catalytic system. This may be a shortcut to highly active catalysts that exceed all conventional catalysts in all temperature ranges, whereby the conventional operating temperature and pressurization are decreased, although the guiding principles to lower the lowest catalyst working temperature have yet to be clarified.

The simply prepared Ru/CaFH presented in this study not only works for ammonia synthesis at

50 °C but exceeds conventional highly active catalysts in activity in all temperature ranges, despite the low pressurization and weight hourly space velocity (Supplementary Tables 2 and 3). In addition, Ru/CaFH also produces ammonia for over 100 h without any decrease in activity even at 340 °C (Supplementary Fig. 8 added to the revised manuscript). The amount of ammonia produced by Ru/CaFH at 340 °C exceeded the amount of the used catalyst (ca. 24 mmol) within 100 min. The amount of Ru/CaFH effective for the reaction (*i.e.*, amount of substrate near the Ru/CaFH surface) is less than several percent of the Ru/CaFH used. Furthermore, there was no significant difference in the XRD patterns and surface F concentrations of Ru/CaFH before and after reaction. These results indicate that Ru/CaFH acts as a highly active and stable catalyst for ammonia synthesis.

According to the reviewer's comment, the following has been revised.

(1) Page 2, lines 23-29 in the original manuscript has been replaced by the following, in accordance with the reviewer's comments.

Page 2, line 23-29.

“Thus, a lower temperature is favorable for ammonia production with respect to yield and energy consumption, and more efficient ammonia production is required to overcome the kinetic barrier at lower temperature to achieve the equilibrium. However, conventional catalysts equally lose the catalytic activity for ammonia formation from N₂-H₂ at 100-200 °C, even if they exhibit high catalytic performance at high temperatures, as shown in Fig. 1a (see below). In addition, guiding principles to lower the temperature for a loss of activity have yet to be clarified,”

(2) “at 50 °C “ (Page 2, line 36 in the original manuscript) has been replaced with “at lower temperatures”.

(3) The physicochemical information (surface area, porosity, Ru particle size) and the rates of ammonia formation for Ru/CaFH, Ru/CaH₂ (comparison catalyst), Ru/BaO-BaH₂, Ru/Ba-Ca(NH₂)₂, Ru/C12A7, and the commercial Fe catalyst and Cs-Ru/MgO (benchmark catalysts) at 100-340 °C are shown as Supplementary Table 2 in the revised manuscript. The catalytic activities of the benchmark catalysts (commercial Fe catalyst and Cs-Ru/MgO) were comparable to those reported by other groups (*Appl. Catal. General*, **142**, 209-222 (1996), *Chem. Sci.* **8**, 674-679 (2017)).

The following has been added to the revised manuscript accordingly with a reference paper (*Appl. Catal. General*, **142**, 209-222 (1996), Reference 20).

Page 6, line 19-25.

“Supplementary Table 2 shows the physicochemical information (surface area, porosity, and Ru particle size) and the rates of ammonia formation for Ru/CaFH, Ru/CaH₂, Ru/Ba-Ca(NH₂)₂, Ru/BaO-BaH₂, Ru/C12A7¹⁹, Cs-Ru/MgO and a commercial Fe catalyst as benchmark catalysts at 100-340 °C. It was confirmed that the catalytic activities of the commercial Fe catalyst and Cs-Ru/MgO benchmark catalysts were comparable to those reported by other groups¹⁹⁻²¹. These conventional catalysts did not exhibit activity for ammonia synthesis below 100-200 °C,”

(4) The rates of ammonia formation and CWYEs for all tested catalysts, Fe-LiH, Co-LiH, BaH₂-Co/CHTs, Ru/La_{0.5}Ce_{0.5}O_{1.75} and Ru/La_{0.5}Pr_{0.5}O_{1.75} at 200-350 °C are summarized in Supplementary Table 3 in the revised manuscript. Fe-LiH, Co-LiH, BaH₂-Co/CHTs, Ru/La_{0.5}Ce_{0.5}O_{1.75} and Ru/La_{0.5}Pr_{0.5}O_{1.75} had been reported as highly active catalysts by other groups.

The following has been added to the revised manuscript, according to the addition of

Supplementary Table 3 in the revised manuscript.

Page 6, line 36- page 7, line 6.

“Table 1 gives the catalyst weight required for the equilibrium yield (CWEY) of ammonia at 200 °C (See Supplementary Section 1.3). The rates of ammonia formation and CWEYs for all tested catalysts, including Ru/Ba-Ca(NH₂)₂, and the recently reported highly active catalysts at 200-350 °C are summarized in Supplementary Table 3²²⁻²⁵. Although Ru/Ba-Ca(NH₂)₂ and Ru/BaO-BaH₂ have had much smaller CWEYs among the reported highly active catalysts, the CWEY of Ru/CaFH was only half that of both catalysts at 200 °C.”

Furthermore, the following has been added to Table 1 and “**1.3. CWEY at each pressure**” of “**1. Additional discussion**” in **Supporting Information**.

“CWEYs were estimated from the rates of ammonia formation at each reaction temperature. It was confirmed that the rates of ammonia formation (μmol h⁻¹) for Ru/CaFH, Ru/BaO-BaH₂ and Ru/CaH₂ increased in direct proportion to the catalyst weight (0.05-5.00 g).”

(5) A time course of the ammonia formation rate for Ru/CaFH at 340 °C has been added to the revised manuscript as Supplementary Fig. 8 with the following caption.

“The amount of ammonia produced by Ru/CaFH at 340 °C exceeded the amount of the used catalyst (ca. 24 mmol) within 100 min.”

Furthermore, Page 6, line 29-page 7, line 1 in the original manuscript has been replaced by the following, according to the addition of Supplementary Fig. 8.

Page 7, line 6-18.

“In addition, Ru/CaFH produced ammonia without a decrease in activity for long periods of time and at higher temperatures (200 and 340 °C) (Supplementary Figs. 7 and 8). Ammonia formation over Ru/CaFH was close to the equilibrium yield, even at ca. 300 °C, because of the high catalytic performance. As a result, ammonia synthesis over Ru/CaFH in Supplementary Fig. 8 reaches the equilibrium. Despite such equilibrium conversion (*i.e.*, catalyst deactivation test conditions), the rate of ammonia formation over Ru/CaFH was constant for over 100 h. The amount of ammonia produced by Ru/CaFH at 340 °C exceeded the amount of the used catalyst (ca. 24 mmol) within 100 min. The XRD pattern and surface atomic ratio of F to Ca (F/Ca = 0.08) for Ru/CaFH were unchanged after reaction for 100 h, which was consistent with the lack of F species such as HF detected during reaction. These results are clearly indicative of the stability of the Ru/CaFH catalyst.”

2. *The relationship between the Ru nanoparticles and ionic support is not discussed. The author used the bulk mechanism to argue the formation of H donator sites and electron donator sites (low work function). The DFT calculations and XRD studies to imply their thermodynamic trend to contribute to Ru is not useful. But, how can these ionic centers move from bulk ionic compound to the Ru surface? If only the surface contact, then some surface studies are required.*

We appreciate the reviewer’s comment for the clarification of the reaction mechanism. We studied the early stage of ammonia synthesis from N₂-D₂ over the catalyst at 180 °C by additional experiments. The results have been added to the revised manuscript as Fig. 3 in the main text. Ru/CaFH prepared in a flow of pure H₂ at 340 °C was cooled down from the temperature to 180 °C in a flow of Ar and held under this flow for 5 h. After each mass signal intensity remained constant, N₂-D₂ was passed into Ru/CaFH at the temperature under atmospheric pressure. First, NH₃ was detected, and then NDH₂ began to form soon after. ND₂H

was observed after NDH₂ generation; ammonia species containing H are produced at the early stage of the reaction. Because H adatoms are not expected to be on the Ru surfaces immediately before the introduction of N₂-D₂, these results imply that H in the CaFH bulk is used for ammonia formation at the early stage of the reaction and the ionic centers can move from the bulk ionic compound to the Ru surfaces.

The following revisions have been made in the revised manuscript.

(1) These experimental results have been added to the main text in the revised manuscript as Fig. 3 “Reaction time profiles for ammonia synthesis from N₂-D₂ over Ru/CaFH at 180 °C”.

(2) The following “Ammonia synthesis from N₂ and D₂” has been added to “**Methods**” in the revised manuscript.

Page 11, line 31-Page 12, line 2.

Ammonia synthesis from N₂ and D₂.

A mixture of 98 mol% CaH₂, 2 mol% modified BaF₂ and Ru(acac)₃ corresponding to 12 wt% Ru was heated in the reactor at 260 °C for 2 h and at 300 °C for 10 h in a flow of H₂ (2.5 mL min⁻¹). The resultant Ru/CaFH was cooled down from that temperature to 180 °C in a flow of Ar at a flow rate of 30 mL min⁻¹. After each mass signal intensity was kept constant, N₂-D₂ (N₂: 5 mL min⁻¹, D₂: 15 mL min⁻¹) was passed into Ru/CaFH at that temperature under atmospheric pressure. The outlet gas from the reactor was analyzed using mass spectrometry (BELMass, MicrotracBEL, Japan).

(3) The following has been added to “**Electron-donating capability and reaction mechanism of Ru/CaFH**” in the revised manuscript.

Page 8, line 2-18.

“Ammonia synthesis from N₂ and D₂ over Ru/CaFH was examined to clarify the reaction mechanism. Ru/CaFH prepared at 340°C in a flow of H₂ alone was cooled down from 340 °C to 180 °C in a flow of Ar, and then N₂-D₂ was passed into Ru/CaFH at temperature under atmospheric pressure (See Methods). The experimental reaction time profiles for ammonia synthesis from N₂-D₂ over Ru/CaFH are shown in Fig. 3. Soon after an increase in the $m/z=17$ signal (NH₃ and NDH as fragments of ND₂H and NDH₂), the signal of $m/z=18$ (NDH₂ and ND₂ as a fragment of ND₃) increased. The $m/z=19$ (ND₂H) signal was observed after ca. 3 min from the introduction of N₂-D₂. The fragment ratio of $m/z=17$ (NH₃), 16 (NH₂) and 15 (NH) in pure NH₃ was ca. 100:80:8, so that each fragment intensity did not exceed the parent intensity. The formation of ND₃ ($m/z=20$) was not observed within 10 min from the beginning of the reaction. As a result, NH₃ was first formed, followed by the formation of NDH₂ and ND₂H; ammonia species containing H was formed at the early stage of ammonia synthesis from N₂-D₂ over Ru/CaFH. These results indicate that H in the CaFH bulk is used for ammonia formation at the early stage of reaction and H can move from CaFH bulk to the Ru surface to react with N adatoms. The same phenomenon had been observed on Ru nanoparticle-deposited Ca₂NH (calcium nitride hydride).¹⁴”

3. H₂-TPD was the only experimental evidence to prove the electron trapping at low temperature. However, the experiment was very roughly performed. The baseline was not even stabilised (Fig. 3 and Fig. S2). The poor signal to noise level cannot be used to claim the onset temperature of H₂ desorption.

Figure 3a and Supplementary Fig. 4 were unclear due to the lack of a background profile with

$m/z=2$ obtained from a blank cell. Each H₂-TPD profile with a background profile was measured again and are shown in Fig. 3a and Supplementary Fig. 4 of the revised manuscript. No desorption of molecular species giving an $m/z=2$ signal, such as H₂O, was observed. There is a clear difference in the H₂-TPD profiles among Ru/CaFH, R/CaH₂ and the blank.

4. *More experiments for example in-situ EPR, XPS et al. are required to support the existence of vacancy and electron trapping.*

According to the reviewer's comment, XPS Ru 3p_{3/2} spectra for Ru/CaFH and Ru-deposited SiO₂ (the average Ru particle size (3.2 nm) was similar to that of Ru/CaFH (3.4 nm)) are shown as Supplementary Fig. 9 in the revised manuscript. The Ru 3p_{3/2} spectra for Ru/CaFH appeared at a slightly lower binding energy than that of Ru/SiO₂, which indicates that Ru on CaFH is more negative than Ru on SiO₂. However, the difference was so small that we could not evaluate the electron donation from CaFH to Ru. This is because XPS reflects only Ru atoms near the edges of Ru particles connected to CaFH with respect to the escape depth of photoelectrons. For this reason, we have adopted FT-IR measurements using N₂ as a probe molecule, a more sensitive method. ν_{N_2} for N₂ adsorbed on SiO₂ was observed at $>2200\text{ cm}^{-1}$, whereas that for Ru/CaFH appears below 2150 cm^{-1} .

The following has been added to “**Electron-donating capability and reaction mechanism for Ru/CaFH**” in the revised manuscript.

Page 8, line 19-26.

“XPS Ru 3p_{3/2} of Ru/CaFH was used to evaluate the electron-donating capability from CaFH to Ru (Supplementary Fig. 9). Ru 3p_{3/2} for Ru/CaFH appeared at a slightly lower binding energy than that for metallic Ru particles deposited on SiO₂ (Ru/SiO₂), which indicates that Ru on CaFH is more negative than Ru on SiO₂. However, the difference was not so large, because XPS reflects only Ru atoms near the edges of Ru particles connected to CaFH from the point of view of the escape depth of photoelectrons. For this reason, we have adopted Fourier transform infrared (FT-IR) spectroscopy measurements using N₂ as a probe molecule, which is a more sensitive method (FT-IR, Fig. 4).”

5. *Quality of SEM images is also poor for statistical counting of particle size.*

According to the reviewer's comment, Supplementary Fig. 6c has been replaced with a clearer STEM image in the revised manuscript.

6. *Catalyst weight required for equilibrium yield (CWEY) was employed to show the activity of catalysts: it is a crude and can easily mislead readers. Many other factors such as catalyst porosity, particle size and support can affect the activity when amplifying the catalysts amount.*

We accept the reviewer's criticism concerning Table 1 and Supplementary Table 2 in the original manuscript. The physicochemical information (surface area, porosity, Ru particle size) and the rates of ammonia formation for Ru/CaFH, Ru/CaH₂ (comparison catalyst), Ru/BaO-BaH₂, Ru/Ba-Ca(NH₂)₂, Ru/C12A7, and a commercial Fe catalyst and Cs-Ru/MgO (benchmark catalysts) at 100-340 °C are shown as Supplementary Table 2 of the revised manuscript. In addition, Supplementary Table 2 in the original manuscript has been replaced with Supplementary Table 3, which includes the rates of ammonia formation and CWEYs for all tested catalysts, Fe-LiH, Co-LiH, BaH₂-Co/CHTs, Ru/La_{0.5}Ce_{0.5}O_{1.75} and Ru/La_{0.5}Pr_{0.5}O_{1.75} at 200-350 °C. Fe-LiH, Co-LiH, BaH₂-Co/CHTs, Ru/La_{0.5}Ce_{0.5}O_{1.75} and Ru/La_{0.5}Pr_{0.5}O_{1.75} had

been reported as highly active catalysts by other groups.

Furthermore, the following has been added to Table 1 and “**1.3. CWEY at each pressure**” of “**1. Additional discussion**” in **Supporting Information**.

“CWEYs were estimated from the rates of ammonia formation at each reaction temperature. It was confirmed that the rates of ammonia formation ($\mu\text{mol h}^{-1}$) for Ru/CaFH, Ru/BaO-BaH₂ and Ru/CaH₂ increased in direct proportion to the catalyst weight (0.05-5.00 g).”

7. *The FT-IR spectra for N₂ adsorption was employed to show the back donation of electron from Ru to N₂. As discussed, the H in the support CaFH can be used for hydrogenation of adsorbed N₂. To illustrate their exchange of atomic/molecular species at both low and high temperature, the authors should label the NH species to demonstrate the interaction between the ionic phase and Ru.*

The FT-IR spectrum noted by the reviewer is an important result in this study. We have added an FT-IR spectrum (1550-1700 cm^{-1}) for N-adsorbed Ru/CaFH at 25 °C to the revised manuscript as Supplementary Fig. 10 with the following.

Page 9, line 3-8.

“Supplementary Fig. 10 shows an FT-IR spectrum (1550-1700 cm^{-1}) for N-adsorbed Ru/CaFH at 25 °C (N₂: 12 kPa) and a band assignable to δNH bending, which can be attributed to adsorbed ammonia from the broad band at ca. 1600 cm^{-1} . This indicates that N₂ molecules are dissociated into N adatoms, which react with H from CaFH even at 25 °C, and is consistent with the H₂-TPD results and ammonia synthesis from N₂-D₂.”

8. *HF may be formed under the reaction conditions, can the CaFH be stable at high temperature?*

Ru/CaFH produced ammonia without a decrease in activity for over 100 h at 340 °C without decrease in activity, and F species such as HF were not detected during the reaction.

A time course of the ammonia formation rate for Ru/CaFH at 340 °C has been added to the revised manuscript as Supplementary Fig. 8 with the following caption.

“The amount of ammonia produced by Ru/CaFH at 340 °C exceeded the amount of the used catalyst (ca. 24 mmol) within 100 min.”

Furthermore, Page 6, line 29-page 7, line 1 in the original manuscript has been replaced with the following.

Page 7, line 6-18.

“In addition, Ru/CaFH produced ammonia without a decrease in activity for long periods of time and at higher temperatures (200 and 340 °C) (Supplementary Figs. 7 and 8). Ammonia formation over Ru/CaFH was close to the equilibrium yield, even at ca. 300 °C, because of the high catalytic performance. As a result, ammonia synthesis over Ru/CaFH in Supplementary Fig. 8 reaches the equilibrium. Despite such equilibrium conversion (*i.e.*, catalyst deactivation test conditions), the rate of ammonia formation over Ru/CaFH was constant for over 100 h. The amount of ammonia produced by Ru/CaFH at 340 °C exceeded the amount of the used catalyst (ca. 24 mmol) within 100 min. The XRD pattern and surface atomic ratio of F to Ca (F/Ca = 0.08) for Ru/CaFH were unchanged after reaction for 100 h, which was consistent with the lack of F species such as HF detected during reaction. These results are clearly indicative of the stability of the Ru/CaFH catalyst.”

9. *There are some format and spelling mistakes.*

Spelling and format errors have been corrected.

Reviewers' Comments:

Reviewer #1:

Remarks to the Author:

This revised version presents improved readability and most of my questions have been responded properly.

Reviewer #2:

Remarks to the Author:

The authors addressed the concerns I raised in my previous report. I have no further concerns with the manuscript.

Reviewer #3:

Remarks to the Author:

I appreciate the efforts of authors in revising the manuscript upon my initial comments. The additional N₂-D₂ and FTIR experiments can help to answer some of my queries. The comparison with explored catalysts from their lab and conventional catalysts at various temperatures in the SI is useful to readers. However, I do not think the authors yet answer my main concern: that is whether this study offers a real innovation in ammonia synthesis as they emphasised in the introduction despite a very low activity was obtained over their catalytic material at low temperatures? The typical activity of rNH₃ over commercial catalysts at their conditions is over 100,000 μmol g⁻¹ h⁻¹ while they talked about 120 μmol g⁻¹ h⁻¹ at 100°C (practical use?). At such low temperatures, surface defects including their suggested one or even other contaminated impurities may contribute towards this activity (as only academic exercise?). I urge the authors to be careful for their temperature/activation barrier studies as the assumption of the same sites cannot simply be projected to work at high temperature or pressure conditions to give more appreciated rates with being taken over by other sites.

Reviewers' comments:

Reviewer #1 (Remarks to the Author):

This revised version presents improved readability and most of my questions have been responded properly.

Response: We are pleased to note the favorable comments of the reviewer.

Reviewer #2 (Remarks to the Author):

The authors addressed the concerns I raised in my previous report. I have no further concerns with the manuscript.

Response: We are pleased to note the favorable comments of the reviewer.

Reviewer #3 (Remarks to the Author):

I appreciate the efforts of authors in revising the manuscript upon my initial comments. The additional N₂-D₂ and FTIR experiments can help to answer some of my queries. The comparison with explored catalysts from their lab and conventional catalysts at various temperatures in the SI is useful to readers. However, I do not think the authors yet answer my main concern: that is whether this study offers a real innovation in ammonia synthesis as they emphasised in the introduction despite a very low activity was obtained over their catalytic material at low temperatures? The typical activity of rNH₃ over commercial catalysts at their conditions is over 100,000 $\mu\text{ mol g}^{-1}\text{ h}^{-1}$ while they talked about 120 $\mu\text{ mol g}^{-1}\text{ h}^{-1}$ at 100°C (practical use ?). At such low temperatures, surface defects including their suggested one or even other contaminated impurities may contribute towards this activity (as only academic exercise ?). I urge the authors to be careful for their temperature/ activation barrier studies as the assumption of the same sites cannot simply be project to work at high temperature or pressure conditions to give more appreciated rates with being taking over by other sites.

Response:

1) As pointed out by the reviewer, different active sites and/or defects may be linked to activity under different conditions. We should have replied to the reviewer's comment in the original manuscript.

In the case of Ru/CaFH, the apparent activation energy ($E_a = 20\text{ kJ mol}^{-1}$) and pre-exponential factor ($A = 10$) in the Arrhenius equation estimated from the rates of ammonia formation at 50, 75, 100 and 125 °C were almost the same as those ($E_a = 23\text{ kJ mol}^{-1}$, $A = 12$) obtained by the ammonia formation rates at 275-340 °C. In addition, there was no significant difference in E_a and A between the reactions over Ru/CaFH under 0.1 and 0.9 MPa. This suggests that the same active sites on Ru/CaFH forms ammonia through a reaction mechanism in a wide range of reaction conditions ($\geq 50\text{ °C}$, $\geq 0.1\text{ MPa}$). It was also confirmed in ammonia synthesis (240-400 °C) over commercial Fe catalyst used in this study that E_a and A under 0.1 MPa are identical with those under 0.9 MPa⁶."

This has been added to the revised manuscript (Page 7, line 9-18).

2)

Figure 1. Schematic activity–temperature curves

Our explanation was not enough in the introduction of the main text, which confused the reviewer. Because conventional catalysts equally lose the catalytic activity for ammonia formation at 100-200 °C as shown in the schematic activity–temperature curves of Figure 1 (an attached file), a further increase in catalytic activity at low temperature range below 300 °C has been significantly limited in conventional approaches. The aim of this study is to lower the temperature for a loss of activity below 50 °C, which largely enhances the catalytic activity for ammonia synthesis at low temperature range below 300 °C (Figure 1). While the rates of ammonia formation at 50 and 100 °C over the resulting Ru/CaFH are not so high, Ru/CaFH surpasses all conventional catalysts in activity over the entire temperature range. This clearly indicates that our new approach can lower the temperature for a loss of activity below 50 °C and results in a catalyst with a blue activity-temperature curve in Figure 1. This has two meanings.

First, the catalytic activities of all conventional catalysts, including commercial Fe catalyst, can be achieved by Ru/CaFH at lower temperatures. This does not necessarily mean that Ru/CaFH can be used at ≤ 100 °C in practical use; Ru/CaFH can remarkably lower the operating temperatures. We reported commercial Fe catalyst to show the ammonia formation rate of 2560 μ mol h⁻¹ g⁻¹ at 400 °C under 0.1 MPa (Reference 6 in the main text). By pressurizing the reaction system to 20 MPa at the temperature, the catalyst shows the commercial ammonia formation rate mentioned by the reviewer. This is natural because catalytic activity increases with increasing pressurization (concentration). On the other hand, Ru/CaFH can exhibit the same ammonia formation rate at ≤ 220 °C (Supplementary Tables 2 and 3). Such a large decrease in reaction temperature cannot be achieved by conventional catalysts.

Second, there is a probability of enhancing the catalytic activity of Ru/CaFH at ≤ 100 °C as conventional approaches have been enhancing the catalytic activity over the temperature for a loss of activity (Figure 1). On the other hand, it is not possible for conventional catalysts to enhance the catalytic activities at ≤ 100 °C because they cannot act at such low temperatures.

To clear the aim of this study, page 2, line 28-32 in the previous main text has been replaced by the following (page 2, line 28-34).

“Lowering the temperature for a loss of activity below 50 °C would largely enhances the catalytic activity for ammonia synthesis at low temperature range below 300 °C. While there has been significant progress in homogeneous catalytic systems to synthesize ammonia from N₂ and H₂ activated by specific and non-reusable reagents below room temperature^{4,5}, guiding principles to lower the temperature for a loss of activity on ammonia synthesis from N₂-H₂ have yet to be clarified.”